# From Recall to Reasoning: The Role of Associative Memory in Hybrid Architectures

## Abstract

The demand for efficient inference has driven the development of subquadratic architectures as alternatives to the Transformer, though their capacity for complex, algorithmic reasoning remains a critical open question. To investigate the effect of architectural choice on downstream reasoning performance, we conduct a controlled study of reasoning scaling laws, training from scratch multiple hybrid-attention architectures of the same size (150M and 500M parameters) across three model classes (Mamba, Gated Linear Attention, Gated Delta Net) on a unified mathematical reasoning curriculum. Furthermore, we apply parallel test-time scaling methods via majority voting, and discover a clear trend that the amount of Attention layers increases reasoning performance. To investigate this trend, we analyze the models' responses using LLM-as-a-Judge and categorize reasoning errors into 8 distinct types inspired by taxonomies in math education, identifying in-context associative recall as the primary error mode in attention-free architectures. As we move toward fully linear models without any attention layers, our findings establish a connection between the choice of architectural update rule and systematic failures. In particular, we find that hybrid models with Gated Delta Net can match and even exceed the performance of pure Transformers on mathematical reasoning. We present a principled empirical study that informs the design and evaluation of next-generation hybrid reasoning models.

## 1 Introduction

The frontier of artificial intelligence is increasingly defined by the capacity for complex, multi-step reasoning. While scaling Transformers has yielded remarkable results, state-of-the-art performance in domains like mathematics and science now hinges on scaling test-time compute: generating and evaluating extensive chains of thought to find a correct solution (Wei et al., 2022; DeepSeek-AI et al., 2025; Snell et al., 2024). This "slow thinking" paradigm, however, collides with the Transformer architecture's $O(N^2)$ complexity, creating a significant efficiency bottleneck (Feng et al., 2025) and motivating the development of subquadratic sequence models.

Architectures based on State Space Models (SSMs) like Mamba (Gu & Dao, 2024; Dao & Gu, 2024) and linear-recurrent variants like Gated Delta Net (Yang et al., 2025b) have emerged as leading alternatives, offering near-linear time complexity. Their strong performance on language modeling benchmarks has fueled optimism that they can serve as drop-in replacements for Transformers. However, a growing body of evidence reveals a persistent "skill gap" (Bick et al., 2025c) on tasks requiring robust in-context recall (Arora et al., 2023). This has led the community to a pragmatic solution: hybrid architectures that interleave subquadratic layers with attention. This approach, however, raises a crucial question: what capability is attention providing that recurrent mechanisms lack? Recent work offers a direct clue: there is a striking dose-response relationship where systematically increasing the proportion of attention layers improves reasoning performance (Chaudhry et al.). This finding strongly suggests that attention provides a fundamental capability that is the very subject of our investigation.

To formalize this tension, we adopt a unifying perspective (Wang et al., 2025c; Sun et al., 2024) which frames sequence models as implementations of a dynamic *associative memory* (Hopfield, 1982). From this viewpoint, the attention mechanism is a powerful, non-parametric memory that stores explicit key-value pairs, while subquadratic models are recurrent systems that compress con-

text into a finite parametric state (a "fast weight" memory) (Schlag et al., 2021). This distinction allows us to introduce our central research question: *Does the memory compression in subquadratic architectures create a fundamental bottleneck for mathematical reasoning?*

To address this question, we conduct a systematic, controlled study of architectural scaling laws for reasoning, focusing on hybrid models and the specific role of attention within them. Whereas prior work (Wang et al., 2025a) performed an empirical study of scaling laws in language modeling, our investigation takes a complementary direction by examining the mechanisms through which attention supports reasoning in hybrid architectures. Instead of relying on distilled models, which can inherit biases from a Transformer teacher (Paliotta et al., 2025; Bick et al., 2025b; Wang et al., 2025b), we train a suite of 150M and 500M parameter models—including a Transformer baseline, Gated Linear Attention, Mamba, and a Gated DeltaNet—on a common curriculum of mathematical text and reasoning traces. We then subject these models to rigorous evaluations through *Parallel Test-Time Scaling*, measuring performance gains from majority voting over multiple sampled solutions.

Our empirical investigation uncovers a "reasoning gap": subquadratic models generally underperform the Transformer model and leverage increases in test-time compute less effectively. To understand the source of this reasoning gap, we employ LLM-as-a-Judge method to classify errors into core mathematical concepts that isolate core reasoning primitives like state tracking and procedural abstraction. While previous studies (Poli et al., 2024) have employed synthetic tasks to probe architectural design, we extend this mechanistic lens to hybrid attention architectures, a rapidly growing class of models. We demonstrate that the reasoning bottleneck is a manifestation of a fundamental architectural trade-off between computational efficiency and memory fidelity. However, we identify that a class of architectures, Gated DeltaNet, that are capable of performing on par and even exceeding Transformers on mathematical reasoning through their special update rule, which we further investigate through detailed ablations. This work provides a principled framework for evaluating sequence models and underscores that robust reasoning in efficient architectures requires explicit solutions to the associative memory gap.

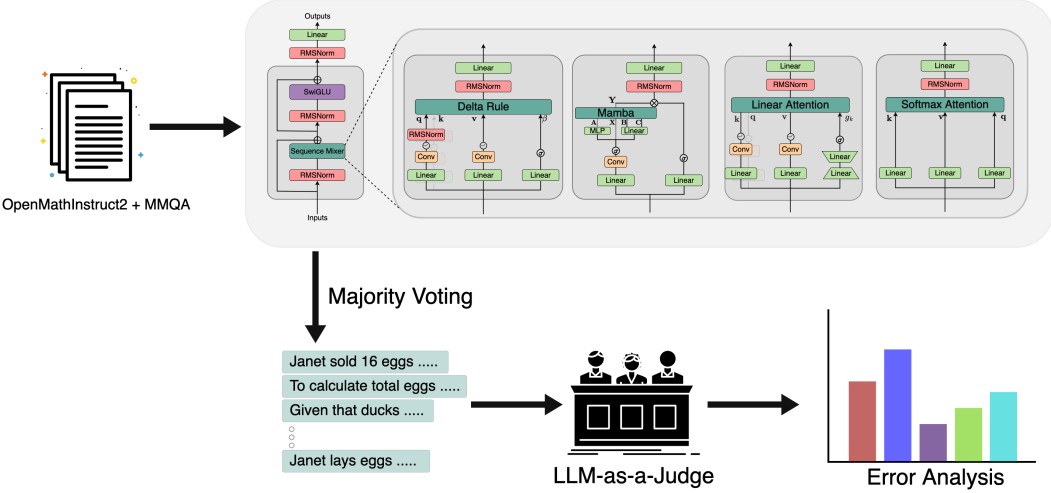

Figure 1: Overview of our experimental pipeline. Models trained on the OpenMathInstruct2 and MMQA datasets incorporate different architectural components (Gated DeltaNet, Mamba, Gated Linear Attention, and Softmax Attention). At test time, predictions are aggregated using majority voting and evaluated via an LLM-as-a-Judge framework, which classifies outputs into distinct error categories to enable fine-grained reasoning analysis.

## 2 RELATED WORK

Our research is situated at the intersection of three active areas: the design of subquadratic architectures, the analysis of their reasoning and memory capabilities, and the scaling of test-time compute.

## 2.1 EXPLOSION OF SUBQUADRATIC ARCHITECTURES

The quadratic complexity of the attention mechanism (Vaswani et al., 2017) has long been a target for optimization. Early efforts focused on linearizing attention, often by recasting it as a recurrent computation (Katharopoulos et al., 2020). This lineage has culminated in a new generation of powerful sequence models based on structured state space models (SSMs). **Mamba** introduced a selective SSM that made parameters data-dependent, dramatically improving performance on discrete data like language (Gu & Dao, 2024). Concurrently, other innovations have focused on improving the recurrent update rule itself, leading to architectures like **Gated Linear Attention (GLA)** (Yang et al., 2024a) and **Gated Delta Network (GDN)**, which incorporate a delta rule for more precise memory modifications (Yang et al., 2024c; 2025b). While these models excel in efficiency, their performance relative to Transformers is a subject of intense study.

## 2.2 SCALING COMPUTE FOR REASONING

The dominant paradigm for improving reasoning in LLMs is to scale compute at inference time. This can be done *sequentially*, by prompting a model to generate longer, more structured chains of thought, often guided by reinforcement learning (DeepSeek-AI et al., 2025) or budgeted thinking (Muennighoff et al., 2025). Alternatively, it can be done in *parallel*, by generating multiple solutions and using a selection mechanism like majority voting (self-consistency) (Wang et al., 2022). Some methods even propose reasoning in a latent space without generating tokens (Hao et al., 2024; Geiping et al., 2025) or apply reinforcement learning at test-time using self-generated rewards (Zuo et al., 2025b). Researchers have recently begun exploring these techniques for subquadratic models, hoping to leverage their higher throughput to outperform Transformers under a fixed time budget (Paliotta et al., 2025; Wang et al., 2025b). These works often rely on *distilling* knowledge from a powerful Transformer teacher (Bick et al., 2025b;a) or linearizing a pre-trained Transformer (Zhang et al., 2025a), which can obscure the inherent capabilities of the subquadratic architecture itself.

In contrast, our work provides a systematic comparison of reasoning with these different model architectures by training models from scratch with the same dataset and systematically measuring their response to increased test-time compute.

## 2.3 ASSOCIATIVE RECALL GAP

Despite their efficiency, a consistent performance gap has been observed between Transformers and subquadratic models on tasks that demand robust in-context learning. Arora et al. (2023) first systematically documented this "recall gap" using an associative recall task. This gap has been framed as a trade-off between a model's state size and its recall ability (Arora et al., 2025), and has been mechanistically explained by the effectiveness of a "Gather-and-Aggregate" mechanism that is more robustly implemented by attention heads (Bick et al., 2025c). This has led to a view of Transformers as powerful associative memory systems (Zhong et al., 2025; Krotov & Hopfield, 2016; Chaudhry et al., 2024), a perspective we take inspirations from. Recent work by Okpekpe & Orvieto (2025a) adds a crucial dimension to this debate, highlighting the role of optimization stability. They demonstrate that subquadratic models like Mamba are far more sensitive to learning rate selection than Transformers on associative recall tasks, suggesting that some of the observed performance gap may be attributable to suboptimal training in addition to architectural limitations. They also uncover distinct scaling behaviors, finding that recurrent models benefit primarily from increased width (hidden state size), whereas Transformers require sufficient depth (at least two layers) to form the "induction head" circuits necessary for robust recall (Olsson et al., 2022).

Our work complements these findings by focusing on the architectural properties themselves. While acknowledging the importance of optimization, we aim to show that even with carefully tuned models, a fundamental performance ceiling exists, which we attribute to the fidelity limits of the underlying associative memory.

## 2.4 HYBRID ARCHITECTURES

The primary response to this performance gap has been the development of hybrid models that seek to balance efficiency and capability by interleaving attention and subquadratic layers (Lieber et al., 2024; Glorioso et al., 2024) or integrating attention in parallel heads (Dong et al., 2024). These

hybrids implicitly concede that attention provides a critical function. Studies have systematically explored these hybrids and confirmed that performance on recall-intensive tasks scales directly with the proportion of attention layers (Wang et al., 2025a; Chaudhry et al.). Other lines of work aim to improve the memory of recurrent models directly through mechanisms like test-time training (Behrouz et al., 2024; 2025a; Oswald et al., 2025; Zhang et al., 2025b) or by framing memory updates within a unified optimization framework (Behrouz et al., 2025b). Our work builds on these insights by explicitly studying the role of attention when reasoning with these hybrid architectures.

## 3 EXPERIMENTAL SETUP

### 3.1 ARCHITECTURE

We pretrain models of approximately 150M parameters, using the open-source OLMo codebase (OLMo et al., 2024). All model consists of 12 layers with 12 heads and a width of 768. The MLP dimension is 8x the model dimension. We use SwiGLU (Shazeer, 2020) and the Llama2 tokenizer (Touvron et al., 2023) with a vocab size of 32,000. We apply RoPE positional encoding (Su et al., 2023) to all self-attention layers and apply no positional encoding to Mamba, GLA and GDN layers. Previous works (Yang et al., 2025a) have shown that positional encodings aren't required since these architectures already represent positional information in its sequential processing. For all linear RNN layers, we apply a short convolution of size 4. For GLA and GDN, we do not apply any output gating and use a d_state of 16 for Mamba models. Appendix A.4 shows various ablation studies regarding specific architectural components of these architectures. For Mamba, we use the implementation from mamba-ssm package (Gu & Dao, 2023). For GDN and GLA, we use the implementations from flash-linear-attention library (Yang & Zhang, 2024; Yang et al., 2023; 2025b).

The pretrained models follow a striped design (Lieber et al., 2024; Glorioso et al., 2024; Ren et al., 2024), where a number of full-attention layers are interleaved in between SSM layers. In the Mamba/GDN/GLA 50 variant, attention is applied in layers 1, 3, 5, 7, 9, and 11. The 75 variant uses layers 3, 7, and 11, while the 83 variant restricts attention to layers 5 and 11. Finally, the 100 variant contains no full-attention layers.

### 3.2 DATASETS

For pretraining, we use a mixture of OpenMathInstruct-2 (Toshniwal et al., 2024) and MetaMathQA (Yu et al., 2024). We adopt this specific composition based on recent findings by Zhao et al. (2025), which demonstrated that this mixture (denoted as OMI2 + MMQA) serves as a highly effective initialization for RL post-training. Their extensive study showed that models pretrained on this combination not only achieve superior pass rates on the MATH-500 benchmark but also exhibit strong positive transfer during the reinforcement learning stage, allowing the model to generalize better to harder mathematical problems.

We train our models for 4 epochs on this mixture, totaling 37.1B tokens. OpenMathInstruct2 (Toshniwal et al., 2024) consists of 14M problem-solution pairs from the GSM8K and MATH500 Datasets generated by Llama-3.1-405B-Instruct (Dubey et al., 2024). MetaMathQA is bootstrapped from GSM8K and MATH500 training datasets and consists of diverse reasoning traces. We do not apply any chat template to the datasets.

### 3.3 TEST TIME SCALING

A model's capacity for reasoning can be effectively measured by its ability to improve performance when allocated more computational resources at inference time. We evaluate this on the widely-used GSM8K (Cobbe et al., 2021) and MATH500 (Hendrycks et al., 2021) benchmarks, using a parallel scaling paradigm. We measure the performance gains from exploring a wide solution space using majority voting. We generate $N$ independent solutions via sampling and report accuracy as a function of $N$, for $N \in \{1, \ldots, 64\}$. This tests the model's ability to converge on a correct answer through diverse reasoning paths. Specific implementation details are provided in Appendix B.4.

## 3.4 LLM-as-a-Judge

LLM-as-a-Judge is an evaluation paradigm in which a capable LLM grades or compares system outputs under an explicit rubric, yielding scalable evaluation with strong judge–human agreement on open-ended tasks (Zheng et al., 2023; Liu et al., 2023; Gu et al., 2024). Foundational studies document both effectiveness and failure modes (e.g., position, verbosity, and self-preference biases) and propose mitigations such as order randomization, rubricized criteria, and multi-judge aggregation (Zheng et al., 2023; Liu et al., 2023; Shi et al., 2024; Tan et al., 2024). Beyond evaluation, judge outputs can be repurposed as training signals (RLAIF) to supervise other models (Bai et al., 2022). In the domain of mathematical reasoning, step-level labels support best-of-$N$ selection and process supervision (PRMs), which have been shown to outperform outcome-only signals (Lightman et al., 2023).

**Evaluation Pipeline.** In our experimental setup, we adapt this paradigm to diagnose the specific cognitive mechanisms driving architectural failure modes. Our pipeline proceeds as follows:

1. **Generation & Test-Time Scaling:** For every problem in the test set (GSM8K and MATH500), we generate $N = 64$ independent solutions using nucleus sampling ($T = 0.8$, $p = 0.9$). We apply majority voting (self-consistency) across these samples to derive the test-time scaling curves.

2. **Sampling for Diagnosis:** Due to the computational cost of judging all $64$ traces per problem, we sample the first $8$ generated solutions for fine-grained error analysis. This provides a representative sample of the model's reasoning diversity.

3. **Structured Adjudication:** We employ Gemini 2.5 Flash-Lite (Google DeepMind, 2025) as the highly performant judge. The judge is provided with the full context: the original question, the ground truth solution, and the generated answer through the prompt outlined in Appendix C.1. Through strict JSON output enforcement, the judge analyzes the answer to determine correctness and isolate reasoning failures.

4. **Judgment Logic and Grouping.** A core challenge in analyzing mathematical reasoning is that a single wrong answer may contain multiple overlapping mistakes. To resolve this, our prompt instructs the judge to identify the single, most critical reasoning error that precipitated the failure. The judge classifies this error into one of 8 distinct categories grounded in mathematics education research (Radatz, 1979; Newman, 1977), as defined in Section 4. We demonstrate some examples for each of the error types in Appendix C.3.

## 4 A Taxonomy of Mathematical Reasoning Errors

While prior work has linked performance gaps to behavioral failures in associative recall (Arora et al., 2023) and optimization instability (Okpekpe & Orvieto, 2025b), the extent to which these deficits impede complex multi-step reasoning remains underexplored. We bridge this gap by systematically classifying the chain-of-thought errors of diverse architectures, identifying which failure modes are mitigated by test-time scaling and increased attention density.

We classify reasoning failures into eight distinct categories, organized by the underlying cognitive deficit. Drawing on error taxonomies from mathematics education (Radatz, 1979; Newman, 1977), we structure our analysis around working memory—a known predictor of mathematical ability (Ashcraft & Kirk, 2001; Raghubar et al., 2010; Geary, 2010). Given that subquadratic models struggle with in-context associative recall, we specifically isolate memory-related failures. This approach aligns with recent findings that educational diagnostic methods effectively transfer to language model evaluation (Mishra et al., 2024).

**In-Context Associative Memory Failures.** These errors represent a failure to construct or maintain a correct internal representation of the problem context. They correspond to failures on basic static and dynamic recall tasks.

1. **Key-Value Binding Error:** A failure at the initial "reading" phase. The model incorrectly extracts a value, hallucinates an entity not present in the text, or swaps values between distinct entities. This results in a flawed problem state before reasoning begins.

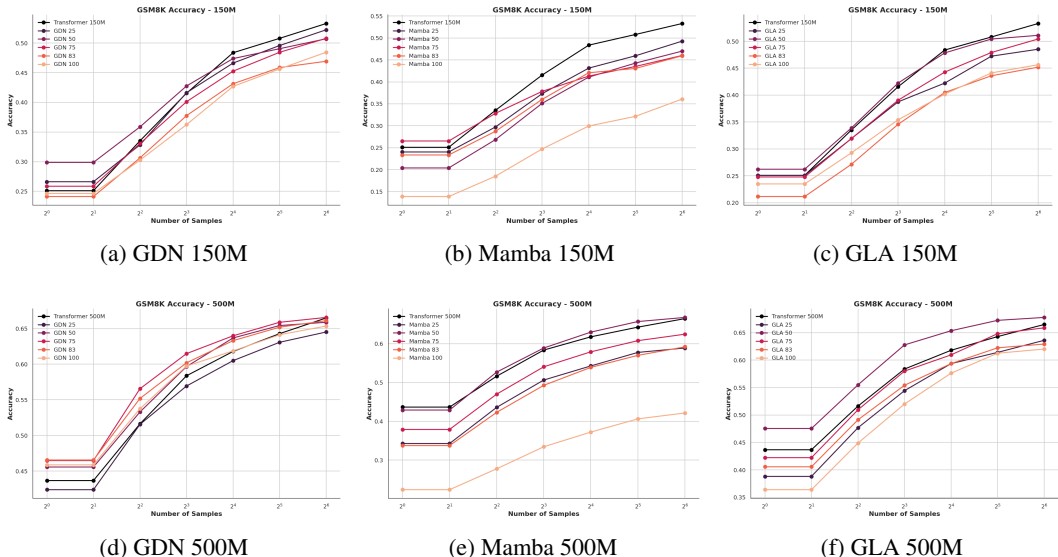

| | | |
|---|---|---|
| (a) GDN 150M | (b) Mamba 150M | (c) GLA 150M |
| (d) GDN 500M | (e) Mamba 500M | (f) GLA 500M |

Figure 2: Parallel Test-Time Scaling via Majority Voting applied across architectures on GSM8K dataset. The Transformer consistently shows the best performance. As the percentage of attention layers increases, the models' performance generally increases and they tend to benefit more than the other models. A clear exception is GDN, which we address in Section 5.2. For 500M models, 50% hybrids perform similar or slightly better than full Transformer.

2. **State Tracking Error:** A failure of dynamic memory update. The model correctly calculates an intermediate value for a changing quantity but fails to propagate this new value to subsequent steps, incorrectly reverting to a stale value.

3. **Context Synthesis Error:** A failure to retrieve the correct set of operands from the internal model during a calculation step. While the internal facts may be correct, the model retrieves an irrelevant distractor value instead of the required one.

**Parametric Memory & Procedural Failures.** These errors occur when the model fails to retrieve or apply general mathematical knowledge stored in its weights, even if the problem context is correctly understood.

4. **Procedural Retrieval Error:** The model applies the wrong algorithm or a flawed version of a standard formula (e.g., using the area formula instead of perimeter, or incorrectly reversing a percentage). This represents a failure of "how-to" knowledge.

5. **Conceptual Knowledge Gap:** The failure stems from a fundamental misunderstanding of an abstract mathematical definition, property, or theorem (e.g., misunderstanding the implication of a "remainder" in a real-world context) rather than a simple formulaic error.

**Logical & Execution Failures.** These errors occur at the highest levels of reasoning, involving abstract planning and final execution, assuming both the internal model of the prompt and procedural knowledge are sound.

6. **Flawed Logical Synthesis:** The model's high-level strategic plan is fundamentally invalid. It connects facts and procedures in a sequence that does not logically address the problem's constraints, often by inventing an unstated goal.

7. **Calculation Error:** A simple arithmetic mistake made during the execution of a valid logical plan. The strategy, procedure, and variables are correct, but a basic computation (e.g., addition or multiplication) is incorrect.

8. **Goal Interpretation Error:** The model executes a valid sequence of steps for a sub-problem but fails to answer the specific final question asked. This often involves reporting an intermediate result as the final answer or solving for a different quantity.

Evaluating model responses against this taxonomy provides a fine-grained understanding of performance gaps on complex benchmarks. Full prompt templates and examples of all eight errors are provided in Appendix C.

## 5 RESULTS

Our experiments reveal a consistent and significant gap in reasoning capabilities between the Transformer and subquadratic architectures. We first present the primary finding on standard benchmarks, demonstrating that subquadratic models fail to effectively leverage test-time compute. We then use our mechanistic analysis tools to diagnose the root cause of this gap, tracing it back to a fundamental deficiency in associative memory, a finding that both builds upon and provides a deeper explanation for the recall gap identified in prior work (Arora et al., 2023).

### 5.1 THE REASONING BOTTLENECK: DIMINISHING RETURNS FROM TEST-TIME COMPUTE

We begin by evaluating our 150M parameter models on the GSM8K benchmark using parallel scaling. As shown in Figure 2, a clear performance hierarchy emerges. The Transformer not only starts with a higher baseline accuracy (pass@1) but also benefits substantially from majority voting, with its performance continuing to climb steeply as the number of samples increases. In contrast, all three subquadratic architectures exhibit a much shallower scaling curve. They show modest initial gains but quickly plateau at a performance ceiling significantly below that of the Transformer. This demonstrates that simply allocating more computational "breadth" at test-time is insufficient to close the reasoning gap. We find similar results on the MATH500 dataset in Figure 4 of Appendix A.

To confirm this finding is not specific to one scaling method or dataset, we conducted further evaluations. When evaluated on the more challenging MATH500 benchmark, the performance delta between the Transformer and subquadratic models becomes less pronounced and even reverses for GDN hybrid models. We hypothesize that this behavior may stem from the distinctive delta-style update rule employed by Gated DeltaNet, which could confer advantages in certain reasoning regimes. These additional results, together with systematic ablations on more hybrid architectures, are presented in Appendix A.

### 5.2 GATED DELTA RULE CLOSES THE REASONING GAP

While our broader analysis in Section 5.1 established that subquadratic hybrids generally plateau earlier than Transformers, we observed a notable exception: the GDN architectures. At both 150M and 500M scale, all the GDN hybrids remained surprisingly competitive with Transformer. To understand the source of this resilience, we investigate the specific mechanics of the GDN update rule. Unlike standard linear attention or Mamba, which primarily employ "Hebbian style" additive updates, GDN incorporates a "delta rule" mechanism. This allows the model to explicitly subtract information from its memory state, effectively implementing a form of online gradient descent on its internal weights at every token step.

For a single layer, the state update rule is formalized as:

$$S_t = \alpha_t S_{t-1} + \beta_t (v_t - S_{t-1} k_t) k_t^\top \tag{1}$$

where $S_t \in \mathbb{R}^{d \times d}$ is the hidden state matrix, and $k_t, v_t \in \mathbb{R}^d$ are the key and value vectors. The term $(v_t - S_{t-1} k_t)$ represents the "prediction error" or the delta between the retrieved value and the target value. Crucially, the update is modulated by two gating factors:

- $\alpha_t \in [0, 1]$: The forget gate (decay) controls how much of the previous state is retained.
- $\beta_t \in \mathbb{R}$: The step size (write gate) controls the magnitude of the correction to the state.

To isolate which component of this mechanism drives reasoning capabilities, we performed a controlled ablation on the 150M parameter model, varying whether $\alpha$ and $\beta$ are a fixed constant ($= 1$), learned scalars, or learned token-dependent projections. The results are presented in Table 1.

The ablations reveal a clear hierarchy: configurations with a token-dependent $\alpha$ consistently outperform those with fixed or scalar decay. Regardless of $\beta$ configuration, constraining $\alpha$ to a learned

Table 1: Ablation of the Gated DeltaNet update rule (150M). We observe that token-dependent decay ($\alpha$) is critical for performance, while the write operation $\beta$ generally seems negligible.

| Configuration | | GSM8K | | MATH500 | |
|---|---|---|---|---|---|
| $\alpha$ (Decay) | $\beta$ (Write) | maj@1 | maj@64 | maj@1 | maj@64 |
| Token-dependent | Token-dependent | 29.0 | 50.0 | **27.2** | 43.0 |
| Token-dependent | Scalar | **29.9** | **50.6** | 25.0 | **44.0** |
| Token-dependent | Constant ($=1$) | 24.5 | 47.9 | 24.2 | 43.4 |
| Scalar | Token-dependent | 23.5 | 46.7 | 24.6 | 43.4 |
| Scalar | Scalar | 26.5 | 46.5 | 26.2 | 43.0 |
| Scalar | Constant ($=1$) | 25.7 | 46.3 | 23.6 | 42.0 |
| Constant ($=1$) | Token-dependent | 21.4 | 47.6 | 23.8 | 39.8 |
| Constant ($=1$) | Scalar | 23.4 | 46.6 | 21.6 | 40.2 |
| Constant ($=1$) | Constant ($=1$) | 23.3 | 46.6 | 22.8 | 40.8 |

scalar or constant value consistently drops performance. For example, simply switching $\alpha$ from a token-dependent to scalar gate (while keeping $\beta$ token-dependent) drop GSM8K maj@1 performance from 29.0% to 23.5%. Interestingly, token-dependent or scalar $\beta$ generally seems to be negligibly better than constant except for when $\alpha$ is token-dependent, in which case changing from scalar or token-dependent $\beta$ to a constant value of 1 significantly drops performance from 29.9% to 24.5%. This also explains why Gated Delta Net significantly outperforms the normal Delta Net architecture (which corresponds in our case to $\alpha = 1$ and token-dependent $\beta$) (Yang et al., 2024b).

This suggests that for mathematical reasoning, which often requires abilities such as discarding stale variable assignments (e.g., $x = 5$) to make room for new ones ($x = 10$), the ability to selectively "forget" via a dynamic $\alpha_t$ is more critical than the precise magnitude of the write operation $\beta_t$.

## 5.3 DIAGNOSING THE REASONING GAP: ASSOCIATIVE MEMORY AS THE CAUSE

Having established the trend of higher Attention percentage generally increasing reasoning performance, we investigate its cause. We hypothesize that the poor performance on mathematical reasoning stems from an underlying failure of in-context associative memory. We test this directly by classifying the reasoning traces of the models into the previously mentioned categories using LLM-as-a-judge. Unlike behavioral tasks such as Multi-Query Associative Recall, which test the holistic skill of recall on synthetic tasks, our probes are meant to unpack associative recall skills specifically in mathematical reasoning. We show the results for GSM8K below and MATH500 in Appendix C.

| | Transformer | Mamba25 | Mamba50 | Mamba75 | Mamba83 | Mamba100 | GLA25 | GLA50 | GLA75 | GLA83 | GLA100 | GDN25 | GDN50 | GDN75 | GDN83 | GDN100 |
|---|---|---|---|---|---|---|---|---|---|---|---|---|---|---|---|---|
| **In-Context Associative Memory Failures** | | | | | | | | | | | | | | | | |
| Key-Value Binding Error | **512.50** | 478.38 | 481.38 | 499.62 | 498.38 | 679.25 | 453.75 | 494.62 | 511.38 | 557.25 | 539.25 | 442.50 | 420.12 | 411.88 | 456.62 | 468.00 |
| State Tracking Error | 22.75 | 11.75 | 24.00 | 27.75 | 25.25 | 12.25 | 10.62 | 22.62 | 26.00 | 25.25 | 25.12 | 13.62 | 26.00 | 29.00 | 28.00 | 32.12 |
| Context Synthesis Error | 93.75 | 89.00 | 111.88 | 112.25 | 123.00 | 68.25 | 83.75 | 109.88 | 103.00 | 108.25 | 116.00 | 89.75 | 113.50 | 124.88 | 120.38 | 113.00 |
| **Parametric Memory & Procedural Failures** | | | | | | | | | | | | | | | | |
| Procedural Retrieval Error | 30.75 | 48.62 | 37.25 | 34.62 | 35.25 | 24.75 | 56.38 | 37.50 | 38.00 | 38.62 | 33.75 | 51.38 | 35.25 | 37.88 | 37.38 | 35.75 |
| Conceptual Knowledge Gap | 4.75 | 8.38 | 5.25 | 5.38 | 7.62 | 3.12 | 9.00 | 5.12 | 7.00 | 5.38 | 7.50 | 8.62 | 9.75 | 6.00 | 6.38 | 7.75 |
| **Logical & Execution Failures** | | | | | | | | | | | | | | | | |
| Flawed Logical Synthesis | 254.38 | 326.25 | 271.62 | 227.88 | 237.38 | 324.50 | 334.62 | 273.62 | 285.62 | 277.38 | 206.75 | 325.12 | 266.50 | 190.62 | 198.75 | 185.38 |
| Calculation Error | 13.75 | 13.50 | 11.25 | 14.62 | 11.50 | 8.38 | 13.50 | 12.12 | 12.25 | 8.25 | 12.62 | 15.50 | 19.25 | 14.12 | 18.25 | 14.12 |
| Goal Interpretation Error | 57.75 | 54.00 | 63.75 | 64.50 | 62.00 | 37.38 | 58.75 | 71.62 | 68.25 | 65.75 | 77.88 | 51.50 | 68.12 | 75.75 | 69.38 | 72.75 |

Table 2: Error Category Decomposition for Model Responses on GSM8K dataset. We take an average across 8 generations per problem in order to account for variance in the LLM's classification. The most common error across each model is the Key-Value Binding Error (bolded).

Note that Key-Value Binding Errors constitute the vast majority of errors chosen by the model, with the second highest category being Flawed Logical Synthesis. This is for GSM8K, which are grade school problems where many of the tasks are word problems where values are associated to variables and the student is meant to do operations on them. For MATH500, which constitutes substantially harder problems that require more mathematical maturity, the predominant error mode is Flawed Logical Synthesis with conceptual knowledge gaps and procedural retrieval errors increasing as shown in the appendix. In Figure 10, we find that the total errors and KV errors tend to decrease with Attention percentage in a similar fashion, with an increase in Attention generally decreasing the

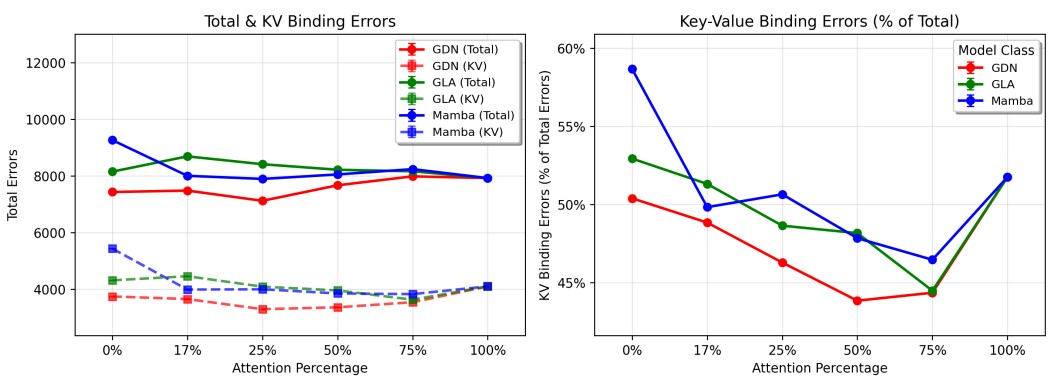

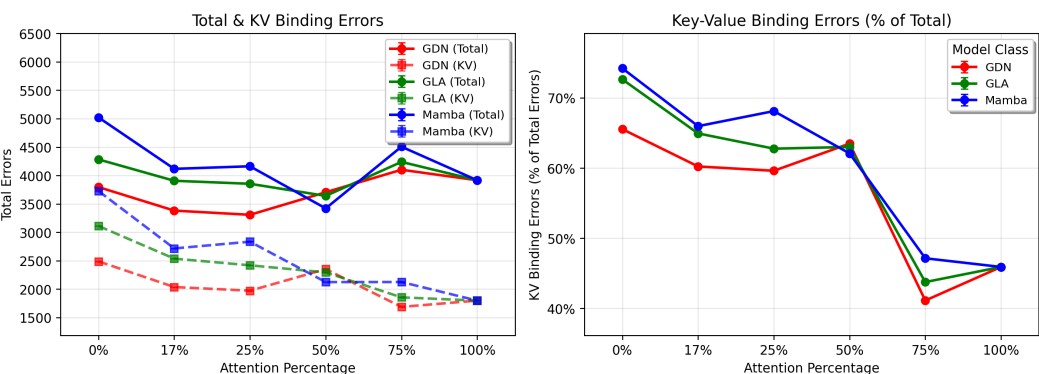

Figure 3: Key–Value binding errors are reduced by attention. **Top:** For 150M models, the total errors and KV-binding errors generally decrease as attention increases across model classes. Note that for the Transformer, there is a slight increase in KV errors as a percentage of the total number of errors, due to a decrease in total errors. **Bottom:** For 500M models, the trend of attention decreasing KV errors is clearer, with a slight jump in the total number of errors at 75% attention due to non KV binding errors.

percentage of KV errors relative to total errors. However, different architectures can have slightly different optimization hyperparameters which can lead to slight deviations from the trend.

## 6 DISCUSSION

Our findings establish a clear link between architectural priors, associative memory deficiency, and the capacity for algorithmic reasoning. The Transformer's non-parametric, token-to-token attention mechanism provides a high-fidelity, content-addressable memory that is resilient to distraction and scales effectively with test-time compute. In contrast, the compressed, fixed-size recurrent state of subquadratic models is a significant bottleneck, leading to rapid memory degradation and a hard ceiling on reasoning performance. Our work provides a strong architectural explanation for the reasoning gap, complementing other recent findings that highlight optimization challenges in sub-quadratic models (Okpekpe & Orvieto, 2025b). Furthermore, we try to account for these difficulties by doing comprehensive hyperparameters sweeps and ablations as shown in Appendix A.

Interestingly, we observe that 50% hybrid architectures can sometimes match or slightly exceed the pass@1 performance of full Transformers (Figure 2, bottom). We hypothesize that this may stem from the mitigation of *rank collapse* or *feature over-smoothing*, a known issue in deep Transformers

where token representations become indistinguishable across layers (Dong et al., 2021). The interleaving of subquadratic layers which employ different mixing mechanisms, likely serves to "reset" the feature space, stabilizing the training signal while allowing the remaining attention layers to focus on high-fidelity retrieval.

This study prompts a critical re-evaluation of what "efficiency" means for reasoning. In Appendix A.2, we show that even when scaling with FLOPs the hybrid models only slightly shift. This means that while subquadratic models are more efficient in FLOPs-per-token, the Transformer makes better use of each computational step, achieving higher accuracy for a given test-time compute budget. This has implications for emerging paradigms like *latent reasoning*, or "Chain-of-Continuous-Thought" (Hao et al., 2024; Geiping et al., 2025), where the success of internal reasoning steps will still hinge on the architecture's ability to maintain a high-fidelity internal state. We anticipate that pure subquadratic architectures would benefit less from these methods than pure Transformers would, although hybrid architectures could strike an interesting balance.

Note that our work focuses on the capabilities of the *internal*, architectural memory of sequence models. An alternative and complementary approach is to augment models with an explicit, *external* memory store. Recent work such as CAMELOT (He et al., 2024) has shown remarkable success with this paradigm, by coupling a frozen LLM with a training-free, consolidated associative memory module to enable the the model to handle arbitrarily long contexts by reading from and writing to this external store. We anticipate that the ideal reasoning architecture would elegantly unify a model with powerful internal associative memory and a persistent, external knowledge store.

These findings regarding the superiority of the Delta rule align with the design choices of recent state-of-the-art hybrid models, such as Qwen3-Next (Team, 2025), and Kimi-Linear (Team et al., 2025). While early hybrids treated attention as a sparse necessity, these large-scale models have independently validated the efficacy of the Gated DeltaNet backbone. Specifically, both Qwen3-Next and Kimi-Linear demonstrate that hybrid architectures dominated by delta-style linear layers (often constituting ~75% of the model depth) can perform competitively with Transformers at scale. Consequently, the "hybrid future" of reasoning likely lies not just in adding attention, but in selecting a recurrent backbone that supports high-fidelity state tracking.

While our experiments focused on mathematical and algorithmic tasks, the requirement for high-fidelity associative memory is a general one. Any domain that requires grounding in long, detailed contexts is likely to be affected by this architectural bottleneck, including long-form question answering, large codebase analysis, and medical or legal document processing.

## 7 CONCLUSION

In this work, we investigated the performance gap between Transformer and hybrid architectures on mathematical reasoning tasks. Through a controlled study of 150M and 500M parameter models trained from scratch, we demonstrated that while all architectures benefit from increased test-time compute, pure subquadratic models often exhibit sharply diminishing returns, hitting a performance ceiling that Transformers easily surpass. We performed error analysis of the reasoning traces using an LLM-as-a-judge rubric targeting primitives underlying mathematical reasoning and found that attention mainly improves performance by increasing memory fidelity and thus decreasing KV binding errors. We conclude that associative memory is not merely one skill among many but a foundational capability upon which robust, scalable reasoning is built. The path toward models that are both efficient and capable reasoners must therefore prioritize the development of architectural priors that explicitly support and preserve memory fidelity.

Our study opens several directions for further research. First, while we validated our findings up to the 500M parameter scale, modern reasoning models typically operate in the multi-billion parameter regime; it remains an open question how the reasoning gap and the efficacy of hybrid ratios evolve at these larger scales. Second, while we carefully tuned optimization settings, it is possible that some of the poor performance of subquadratic models arises from sensitivity to training dynamics rather than fundamental architectural limitations. Finally, our results highlight the relative strength of Gated DeltaNet, whose delta-style update rule enables more effective use of finite recurrent memory. We believe this mechanism deserves greater focus, both as a promising architectural direction in its own right and as inspiration for designing more precise and efficient memory update rules.

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

# A    ADDITIONAL RESULTS

This section provides supplementary results that expand upon the core findings presented in the main body of the paper, including performance on more challenging benchmarks, alternative scaling methods, and architectural ablations.

## A.1    PERFORMANCE ON THE MATH500 BENCHMARK

To validate that our findings generalize to more complex mathematical reasoning, we evaluated all 150M parameter models on the MATH500 benchmark (Hendrycks et al., 2021). As shown in Figure 4, the performance gap between the Transformer and subquadratic architectures is less pronounced on this more challenging dataset. The Transformer demonstrates a significant advantage in maj@64 performance for all Mamba and GLA models, but performs similar than GDN hybrid models.

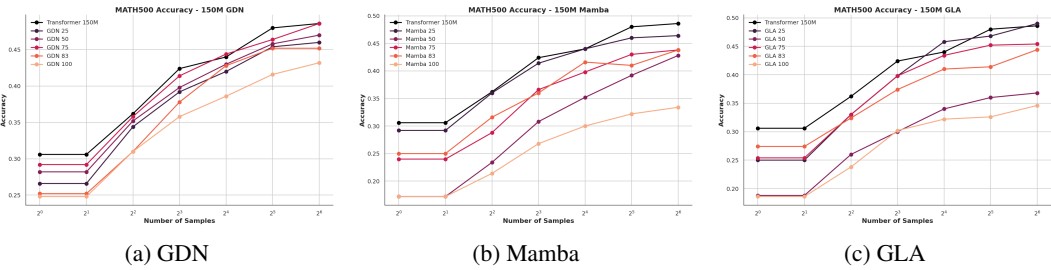

| (a) GDN | (b) Mamba | (c) GLA |

Figure 4: Test-Time Scaling across architectures on MATH500. We use a generation length of 2048.

## A.2    PERFORMANCE ON GSM8K WITH RESPECT TO INFERENCE FLOPs

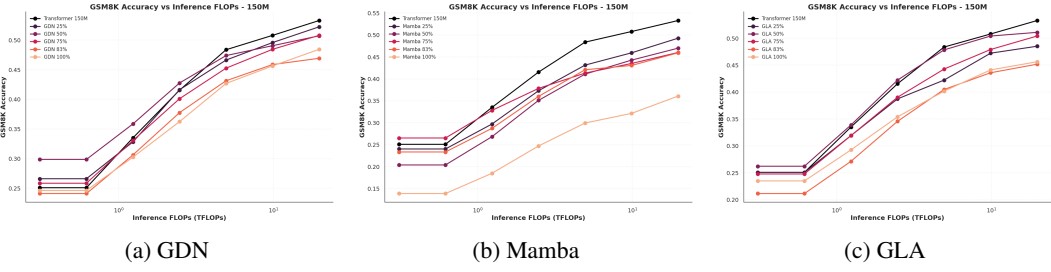

| (a) GDN | (b) Mamba | (c) GLA |

Figure 5: Parallel Test-Time Scaling via Majority Voting with respect to FLOPs applied across architectures on GSM8K dataset. We get similar performance relative to the original. FLOPs are estimated using conversion values from (Wang et al., 2025a).

## A.3    PASS@K (COVERAGE) PERFORMANCE ON MATH500 BENCHMARK

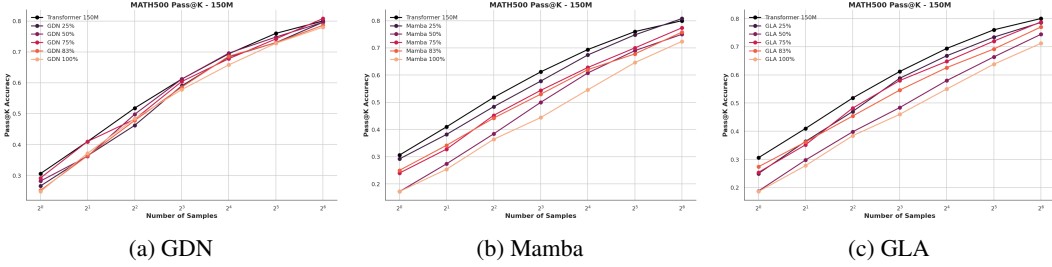

| (a) GDN | (b) Mamba | (c) GLA |

Figure 6: Pass@k scaling for hybrid models. GDN hybrids provides competitive coverage to transformers, while Mamba and GLA lag slightly.

### A.4 ABLATION STUDY: EFFECT OF ARCHITECTURAL COMPONENTS

Mamba based models have shown to be hard to optimize with the optimal learning rate playing a critical role in the performance (Okpekpe & Orvieto, 2025b). We conduct ablation studies on various architectural primitives like ShortConv, dt_rank, and attention placement in our 150M hybrid models.

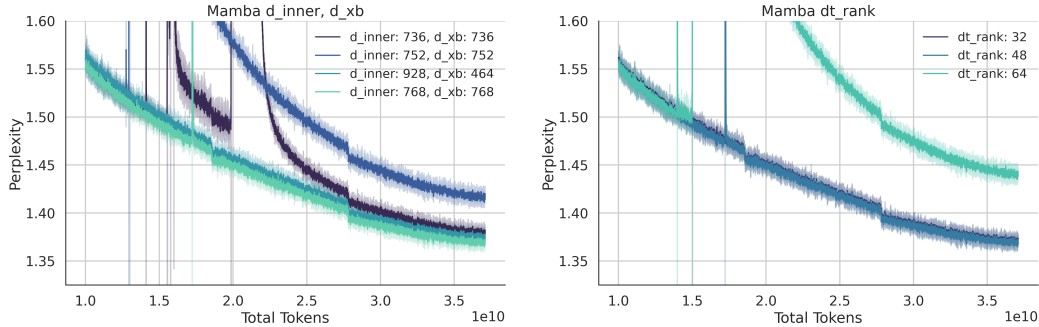

Figure 7: For mamba models, (**Right**) we sweep over some reasonable choices of $d_{inner}$ and $d_{xb}$. Note that these choices are independent of the model dimension (768). (**Left**) To understand why spikes occur in Mamba models, we tried changing the dt_rank parameter. Intuitively this controls the information written to the hidden state. We find conservative values of dt_rank reduce the instabilities. Related, Zuo et al. (2025a) reported that clipping dt to positive values also help with instabilities.

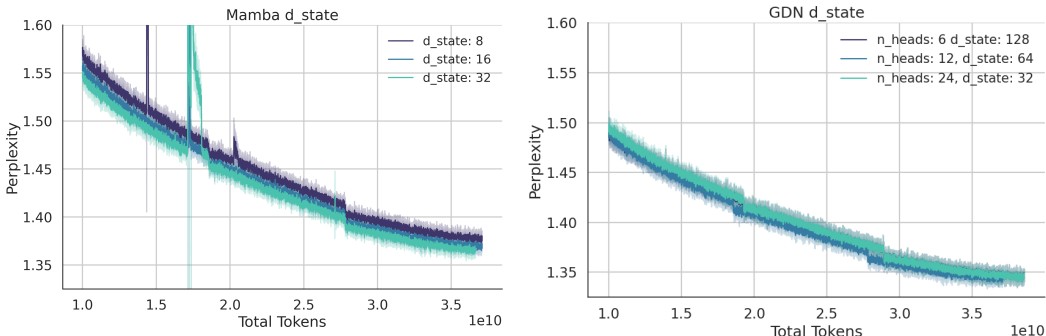

Figure 8: (**Left**) Mamba100 state size. We encounter training instabilities for high state sizes. (**Right**) GDN100: we vary state size, while simultaneously varying $n_{heads}$ to keep the model dimension constant.

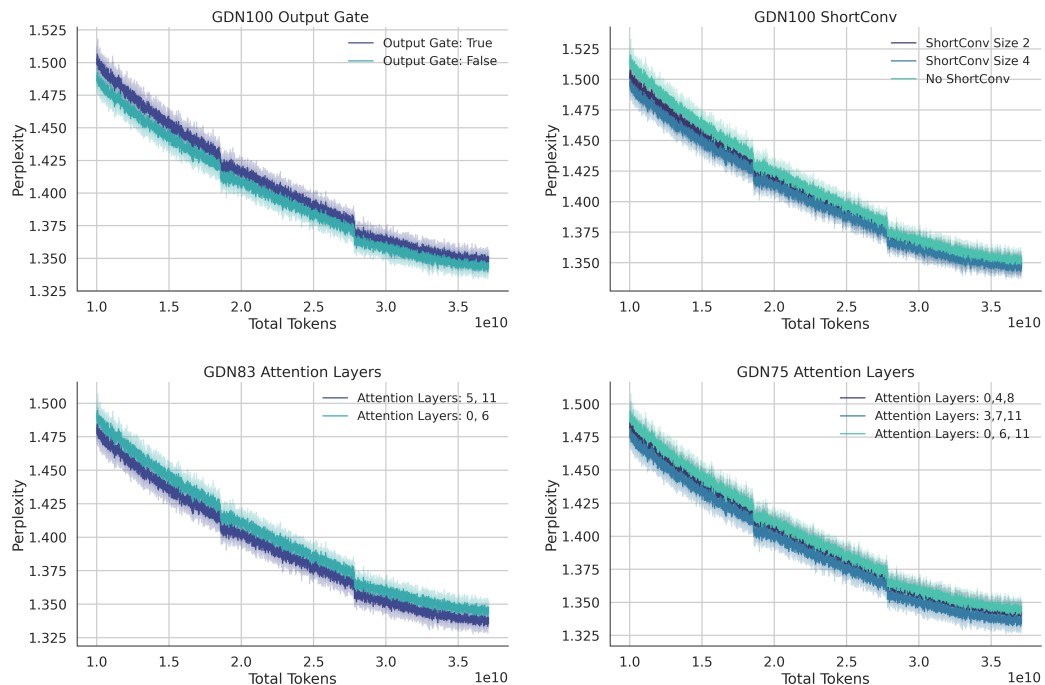

Figure 9: (**Top Left**) Sweep over the Output Gate. (**Top Right**) ShortConv is an important piece in the architectural backbone of Gated DeltaNet. While the size of the convolution does not matter, its existence is important to retain sufficient reasoning capabilities. (**Bottom Left**) For Gated DeltaNet 83, we try various permutations of inserting attention Layers. Attention in the last layer (11) greatly improves performance, also seen in (**Bottom Right**) for GDN73.

Table 3: Hyperparameters for 150M and 500M models

| Parameters | 150M | 500M |
|---|---|---|
| Model dimension | 768 | 1536 |
| Number of Layers | 12 | |
| Number of H"eads | 12 | |
| Number of Key-Value Heads | 12 | |
| MLP Ratio | 8 | |
| Max Sequence length | 2048 | |
| Embedding Size | 32000 | |
| Activation Type | SwiGLU | |
| Positional Embedding | RoPE | |
| RoPE Theta | 10000 | |
| Optimizer Type | AdamW | |
| Adam Betas | (0.9, 0.95) | |
| Learning Rate | 0.001 | |
| Weight Decay | 0.1 | |
| Tokenizer | Llama-2-7b | |

# B  ADDITIONAL EXPERIMENTAL DETAILS

This section provides a comprehensive overview of the models, datasets, and training configurations used in our study, ensuring full reproducibility.

## B.1  MODEL ARCHITECTURES

All four models were implemented in the OLMo codebase (OLMo et al., 2024) using PyTorch and open source triton implementation from Flash-Linear-Attention (Yang & Zhang, 2024). The key architectural hyperparameters for each type of layer for 150M models are detailed in Tables 3

| Hyperparameter | Mamba |
|---|---|
| d_state | 16 |
| d_conv | 4 |
| expand | 1 |
| d_xb | 768 |
| d_inner | 768 |
| dt_rank | 32 |
| **Total Parameters** | **12.1M** |

Table 4: Mamba configuration

| Hyperparameter | GDN |
|---|---|
| expand_v | 1.0 |
| expand_k | 1.0 |
| use_gate | false |
| use_short_conv | true |
| conv_size | 4 |
| conv_bias | false |
| use_qk_norm | true |
| **Total Parameters** | **11.5M** |

Table 5: GDN configuration

| Hyperparameter | GLA |
|---|---|
| expand_k | 1.0 |
| expand_v | 1.0 |
| use_short_conv | true |
| conv_size | 4 |
| conv_bias | false |
| use_output_gate | false |
| gate_fn | swish |
| feature_map | null |
| gate_logit_normalizer | 16 |
| gate_low_rank_dim | 16 |
| clamp_min | null |
| fuse_norm | true |
| **Total Parameters** | **11.5M** |

Table 6: GLA configuration

## B.2  TRAINING DATASETS AND CURRICULUM

Our training curriculum was designed to build broad mathematical competency specializing in reasoning.

**Dataset Composition.** The initial supervised fine-tuning dataset was a blend of the following open-source resources:

- **OMI2:** OpenMathInstruct (Toshniwal et al., 2024) consists of 14M question-answer pairs. The dataset was constructed by prompting Llama3.1-405B to 1) Generate solutions for GSM8K and MATH500, and 2) Create new question-answer pairs similar to the original datasets.
- **MMQA:** MetaMathQA (Yu et al., 2024) is generated using a novel bootstrapping method where the question answer diversity is prioritized. The diversity is particularly important in reasoning directions,

The final mixture consisted of a 1:1 ratio of OpenMathInstruct and MetaMathQA for a total of 9.3B tokens. We train our models on 4 epochs of this dataset, resulting in total 37.1B tokens.

## B.3 TRAINING HYPERPARAMETERS

We use the AdamW optimizer (Kingma & Ba, 2017; Loshchilov & Hutter, 2019) for all of our models with a learning rate of 1e-3 and a weight decay of 0.1. We use a cosine decay scheduler to 10% of the peak learning rate and a linear warmup of 5000 steps.

Key hyperparameters were kept consistent across all architectures and are detailed in Table 7.

Table 7: Training Hyperparameters.

| Hyperparameter | Value |
|---|---|
| **Optimizer** | |
| Name | AdamW |
| Learning Rate | 0.001 |
| Weight Decay | 0.1 |
| Epsilon (eps) | 1e-8 |
| Decay Norm and Bias | true |
| Decay Embeddings | true |
| **Scheduler** | |
| Name | cosine_with_warmup |
| Warmup Steps (t_warmup) | 5000 |
| Final LR Ratio ($\alpha_f$) | 0.1 |
| Warmup Min LR | 0 |
| **Tokenizer** | |
| Identifier | meta-llama/Llama-2-7b-hf |

## B.4 TEST-TIME SCALING IMPLEMENTATION

**Parallel Scaling (Majority Voting).** For all majority voting experiments, we used nucleus sampling with a temperature of 0.8 and a top-p value of 0.9 for a generation length of 1024 for GSM8K and 2048 for MATH500. The final numerical answer was extracted from each of the $N$ generated outputs using the Math_Verify library from HuggingFace.

# C  ERROR ANALYSIS

## C.1  LLM-AS-A-JUDGE PROMPT

---

**Prompt**

You are an expert AI diagnostician specializing in mathematical reasoning errors. Your task is to analyze a single generated answer and classify its primary error if it is incorrect.

— Context —

Question: {question}

Correct Answer's Final Solution: {correct_answer}

— Task —

Analyze the following generated answer:

Generated Answer: "{generated_answer}"

— Instructions —

1. First, determine if the final boxed answer in the "Generated Answer" is correct.

2. If the answer is **CORRECT**, classify the error as **"No Error"**.

3. If the answer is **INCORRECT**, identify the single, most critical reasoning error and classify it into **exactly one** of the following 8 categories based on the detailed definitions below.

— Error Taxonomy —

**Group A: In-Context Associative Memory (ICAM) Failures (Errors in creating and using an internal model of the problem from the prompt text)**

1. Key-Value Binding Error: A failure at the initial "reading" phase. The model incorrectly extracts a value from the text, hallucinates a value or entity not present, or swaps values between two distinct entities. This results in a flawed internal set of facts before reasoning begins.

2. State Tracking Error: A failure of dynamic memory update. The model correctly calculates an intermediate value for a changing quantity but then fails to use this new value in a subsequent step, incorrectly reverting to a stale (old) value.

3. Context Synthesis Error: A failure during a calculation step to retrieve the correct set of values from its internal model of the problem. The model's internal facts are correct, but it incorrectly gathers them, often retrieving an irrelevant distractor number instead of the required value for a specific operation.

**Group B: Parametric Memory & Procedural Failures (Errors in recalling general knowledge from weights)**

4. Procedural Retrieval Error: The model retrieves the wrong algorithm or a systematically flawed ("buggy") version of the correct one. The error is in the "how-to" knowledge for a standard mathematical process, like using the formula for area instead of perimeter, or incorrectly reversing a percentage.

5. Conceptual Knowledge Gap: The failure stems from a misunderstanding of an abstract mathematical definition, property, or theorem. It is not just a wrong formula, but a deeper lack of understanding of the principles governing the problem (e.g., what a "remainder" implies in a real-world context, or the definition of a prime number).

**Group C: Logical & Execution Failures (Errors in high-level planning and final execution)**

6. Flawed Logical Synthesis: The model's high-level strategic plan is fundamentally invalid or nonsensical from the start. It connects facts and procedures in a sequence that does not logically address the problem's context or constraints, often by inventing an unstated goal.

7. Calculation Error: A simple arithmetic mistake made during the execution of an otherwise correct and logical plan. The strategy, procedure, and all variables are correct, but a basic computation (e.g., addition, multiplication) is wrong.

8. Goal Interpretation Error: The model executes a valid and logical sequence of steps for a sub-problem but fails to answer the specific, final question asked. This often involves stopping short and reporting an intermediate result as the final answer or solving for a different quantity altogether.

Return ONLY a single, valid JSON object with this exact structure:

{

"is_correct": true/false,

"error_category": "Category Name" or "No Error",

"error_group": "A", "B", "C", "None", or "Error",

"explanation": "A brief, one-sentence explanation of the error or correctness.",

"erroneous_step_quote": "The exact quote from the generated answer where the primary mistake occurs." or null

}

---

## C.2  ILLUSTRATIVE ERROR TYPES

Below are concise examples for each of the eight refined error categories. Each includes the problem context, the generated answer, and an explanation highlighting the error.

## Key-Value Binding Error

**Context:** Janet's ducks lay 16 eggs per day. She eats 3 for breakfast and uses 4 for baking. She sells the remainder at $2 each. How much does she make?

**Generated Answer:** "Janet's ducks lay 61 eggs per day. She uses 3 + 4 = 7 eggs. This leaves 54 eggs to sell. 54 * $2 = $108. The answer is 108."

**Explanation:** The model misread the initial fact: 16 eggs became 61. All subsequent steps are logically valid, but because the first entity-value binding was wrong, the entire solution is invalid.

## State Tracking Error

**Context:** Janet's ducks lay 16 eggs per day. She eats 3 for breakfast and uses 4 for baking. She sells the remainder at $2 each.

**Generated Answer:** "Janet has 16 eggs. She eats 3, leaving 13. Then she sells these 13 for $2 each. 13 * 2 = 26. The answer is 26."

**Explanation:** The model performed the first update ($16 \rightarrow 13$) but forgot the second update (-4 for baking). It failed to track the evolving state of the eggs.

## Context Synthesis Error

**Context:** Janet's ducks lay 16 eggs per day. She eats 3 for breakfast and uses 4 for baking. She sells the remainder at $2 each.

**Generated Answer:** "Janet has 9 eggs left. She sells them at the same price as the number of eggs she eats for breakfast: $3. 9 * 3 = $27. The answer is 27."

**Explanation:** The model retrieved the wrong value (3, from breakfast eggs) instead of the actual price ($2). The facts exist in memory, but the wrong one was synthesized into the calculation.

## Procedural Retrieval Error

**Context:** A rectangle has length 9 and width 2. Find the perimeter.

**Generated Answer:** "The perimeter is length times width. 9 * 2 = 18. The answer is 18."

**Explanation:** The model recalled the procedure for area instead of perimeter (2L + 2W). This is a retrieval failure of the correct formula.

## Conceptual Knowledge Gap

**Context:** 16 people need cars that hold 3 people each. How many cars are needed?

**Generated Answer:** "16 ÷ 3 = 5 remainder 1. So they need 5 cars. The answer is 5."

**Explanation:** The procedure (division) is fine, but the model lacks conceptual understanding: the remainder implies an extra car is necessary. The correct answer is 6.

## Flawed Logical Synthesis

**Context:** Janet's ducks lay 16 eggs per day. She eats 3 for breakfast and uses 4 for baking. She sells the remainder at $2 each.

**Generated Answer:** "Janet gives 4 muffins to friends. If she sold them for $2 each, she would make 8. The answer is 8."

**Explanation:** The model fabricated an "opportunity cost" problem not stated in the question. The plan itself is logically consistent but irrelevant to the actual goal.

### Calculation Error

**Context:** Janet's ducks lay 16 eggs per day. She eats 3 for breakfast and uses 4 for baking. She sells the remainder at $2 each.
**Generated Answer:** "Janet uses 3 + 4 = 8 eggs. She has 16 - 8 = 8 left. She makes 8 * $2 = $16. The answer is 16."
**Explanation:** Everything is conceptually correct, but a simple addition mistake (3+4=8 instead of 7) corrupted the result.

### Goal Interpretation Error

**Context:** Janet's ducks lay 16 eggs per day. She eats 3 for breakfast and uses 4 for baking. She sells the remainder at $2 each.
**Generated Answer:** "Janet uses 3 + 4 = 7 eggs. Remaining = 16 - 7 = 9. The answer is 9."
**Explanation:** The plan is valid but incomplete: the model stopped at the number of eggs left, instead of completing the final step (selling them for money).

## C.3   REAL ERROR EXAMPLES

In this section, we present actual error traces generated by the GLA100-500M parameter model during evaluation. These examples illustrate how the specific error definitions defined in Section C.2 manifest in real model outputs. We specifically showcase how different cognitive failures can occur even on the same problem (Problem 34 from MATH), as well as other illustrative examples.

### Key-Value Binding Error

**Question:** Find the constant term in the expansion of $\left(10x^3 - \frac{1}{2x^2}\right)^5$
**Generated Answer:** "The general term in the expansion of $(a + bx^m)^n$ is $\binom{n}{r}a^{n-r}(bx^m)^r$. Here, $a = 10x^3$, $b = -\frac{1}{2x^2}$, $n = 5$, and $m = 3$, $r = 0$."
**Erroneous Step:** "Here, $a = 10x^3$, $b = -\frac{1}{2x^2}$, $n = 5$, and $m = 3$, $r = 0$."
**Explanation:** The model incorrectly identifies 'a' as '$10x^3$' and 'b' as '$-1/(2x^2)$' in the general term formula, when 'a' should be the constant part of the first term and 'b' the coefficient of the second term's variable part.

### Procedural Retrieval Error

**Question:** Find the constant term in the expansion of $\left(10x^3 - \frac{1}{2x^2}\right)^5$
**Generated Answer:** "Using the binomial theorem, the term that does not contain any variable is obtained when three of the terms contribute $10x^3$ and the other two contribute $-\frac{1}{2x^2}$."
**Erroneous Step:** "Using the binomial theorem, the term that does not contain any variable is obtained when three of the terms contribute $10x^3$ and the other two contribute $-\frac{1}{2x^2}$."
**Explanation:** The model correctly identifies the need for the Binomial Theorem but retrieves a flawed procedure for determining the cancellation of terms. It incorrectly states that 3 terms of $10x^3$ and 2 terms of $-\frac{1}{2x^2}$ will yield a constant, whereas the correct combination is 2 and 3 respectively.

**Conceptual Knowledge Gap**

**Question:** Find the constant term in the expansion of $\left(10x^3 - \frac{1}{2x^2}\right)^5$

**Generated Answer:** "However, the constant term must not have any $x$ in it... But, there is no $x^0$ term in the expansion, which means that the constant term in the expansion of $\left(10x^3 - \frac{1}{2x^2}\right)^5$ is actually $\boxed{0}$."

**Erroneous Step:** "However, there is no $x^0$ term in the expansion, which means that the constant term in the expansion of $\left(10x^3 - \frac{1}{2x^2}\right)^5$ is actually $\boxed{0}$."

**Explanation:** The model incorrectly concludes that the constant term is 0 because it believes there is "no $x^0$ term," despite correctly calculating that the powers of $x$ cancel out. It fails to understand the concept that $x^0 = 1$ and thus the term containing it is the constant term.

---

**Flawed Logical Synthesis**

**Question:** Find the constant term in the expansion of $\left(10x^3 - \frac{1}{2x^2}\right)^5$

**Generated Answer:** "This can happen in two ways: 1. The term $(10x^3)^5$ is the constant term. 2. The term $\left(-\frac{1}{2x^2}\right)^5$ is the constant term."

**Erroneous Step:** "This can happen in two ways: 1. The term $(10x^3)^5$ is the constant term. 2. The term $\left(-\frac{1}{2x^2}\right)^5$ is the constant term."

**Explanation:** The model's high-level strategic plan is fundamentally invalid. It incorrectly assumes that the constant term can only arise from the individual terms raised to the power of 5, completely ignoring the binomial expansion's structure where terms are products of two components.

---

**Goal Interpretation Error**

**Question:** Find the constant term in the expansion of $\left(10x^3 - \frac{1}{2x^2}\right)^5$

**Generated Answer:** "However, since the problem asks for the constant term of the expansion, and the constant term does not have any variable part, this term is independent of the constant term. Thus, the constant term is simply the term that does not depend on $x$, which is $\boxed{0}$."

**Erroneous Step:** "However, since the problem asks for the constant term of the expansion, and the constant term does not have any variable part, this term is independent of the constant term. Thus, the constant term is simply the term that does not depend on $x$, which is 0."

**Explanation:** The model correctly identifies that the constant term is independent of $x$, but then incorrectly concludes that the constant term is 0 without calculating the actual constant value. It stops short of the actual calculation.

---

**Calculation Error**

**Question:** How many positive whole-number divisors does 196 have?

**Generated Answer:** "In our case:

$$(a_1 + 1) \times (a_2 + 1) \times \cdots \times (a_n + 1) = (2 + 1) \times (2 + 1) \times (7 + 1) = 3 \times 3 \times 8 = \boxed{72}$$

"

**Erroneous Step:** "$3 \times 3 \times 8 = \boxed{72}$"

**Explanation:** The calculation incorrectly includes the prime factor 7 in the multiplication for the number of divisors (adding a term for the base rather than just the exponents), which leads to the wrong result despite the correct procedure.

---

**Context Synthesis Error**

**Question:** Bill walks $\frac{1}{2}$ mile south, then $\frac{3}{4}$ mile east, and finally $\frac{1}{2}$ mile south. How many miles is he, in a direct line, from his starting point?

**Generated Answer:** "The difference in southward direction: $\frac{1}{2} - \frac{1}{2} = 0$ miles"

**Erroneous Step:** "The difference in southward direction: $\frac{1}{2} - \frac{1}{2} = 0$ miles"

**Explanation:** The model correctly identifies the values ($\frac{1}{2}$ south and another $\frac{1}{2}$ south) but incorrectly synthesizes the context by calculating the total displacement as the difference between the two south movements instead of their sum.

---

**State Tracking Error**

**Question:** A bookcase has 3 shelves... Top shelf has 8 mystery books... Two books are now taken off each shelf. What fraction of the books remaining...

**Generated Answer:** "1. Top shelf: 24 - 8 = 16 books remaining"

**Erroneous Step:** "1. Top shelf: 24 - 8 = 16 books remaining"

**Explanation:** The model correctly calculates the updates for other shelves but fails to track the state of the top shelf. It subtracts the initial number of mystery books from the total number of books (24 - 8) rather than subtracting the removed books from the specific shelf count (8 - 2).

---

## C.4 ERROR ANALYSIS ON MATH500

| | Transformer | Mamba25 | Mamba50 | Mamba75 | Mamba83 | Mamba100 | GLA25 | GLA50 | GLA75 | GLA83 | GLA100 | GDN25 | GDN50 | GDN75 | GDN83 | GDN100 |
|---|---|---|---|---|---|---|---|---|---|---|---|---|---|---|---|---|
| **In-Context Associative Memory Failures** | | | | | | | | | | | | | | | | |
| Key-Value Binding Error | 51.62 | 35.62 | 52.12 | 46.00 | 57.62 | 75.50 | 34.00 | 59.62 | 60.88 | 63.50 | 71.00 | 38.38 | 41.88 | 44.75 | 47.88 | 52.38 |
| State Tracking Error | 2.00 | 0.50 | 1.62 | 1.75 | 1.88 | 0.38 | 0.38 | 1.12 | 1.38 | 1.38 | 2.25 | 0.50 | 1.75 | 1.62 | 1.75 | 1.62 |
| Context Synthesis Error | 10.25 | 7.25 | 8.25 | 12.00 | 14.12 | 10.25 | 6.38 | 8.88 | 9.88 | 12.62 | 11.75 | 8.00 | 11.50 | 10.00 | 12.25 | 10.12 |
| **Parametric Memory & Procedural Failures** | | | | | | | | | | | | | | | | |
| Procedural Retrieval Error | 57.38 | **78.12** | 62.12 | 59.62 | 59.12 | 63.12 | **76.88** | 66.00 | 70.88 | 65.88 | 64.62 | **75.88** | 67.12 | 59.50 | 61.62 | 56.00 |
| Conceptual Knowledge Gap | 48.00 | 55.88 | 47.75 | 47.00 | 46.62 | 40.50 | 52.62 | 49.50 | 49.62 | 48.62 | 48.12 | 54.75 | 43.88 | 47.62 | 52.88 | 49.75 |
| **Logical & Execution Failures** | | | | | | | | | | | | | | | | |
| Flawed Logical Synthesis | **103.50** | 78.12 | 100.12 | 92.88 | 92.62 | **112.88** | 74.75 | **112.38** | 113.12 | **114.88** | 96.12 | 74.25 | **105.62** | 80.12 | **85.62** | **85.00** |
| Calculation Error | 35.38 | 27.25 | 36.50 | 37.25 | 35.00 | 25.38 | 32.88 | 29.00 | 27.12 | 27.25 | 31.25 | 29.50 | 31.88 | 40.50 | 41.25 | 40.62 |
| Goal Interpretation Error | 32.75 | 32.25 | 36.12 | 33.88 | 28.62 | 34.00 | 34.00 | 39.50 | 38.50 | 34.12 | 37.50 | 34.25 | 30.75 | 32.25 | 32.38 | 33.12 |

Table 8: Error Category Decomposition for Model Responses on the MATH500 dataset. Means are averaged across 8 generations per problem. The largest value per column is in **bold**, which generally tends to be Flawed Logical Synthesis or Procedural Retrieval Error.

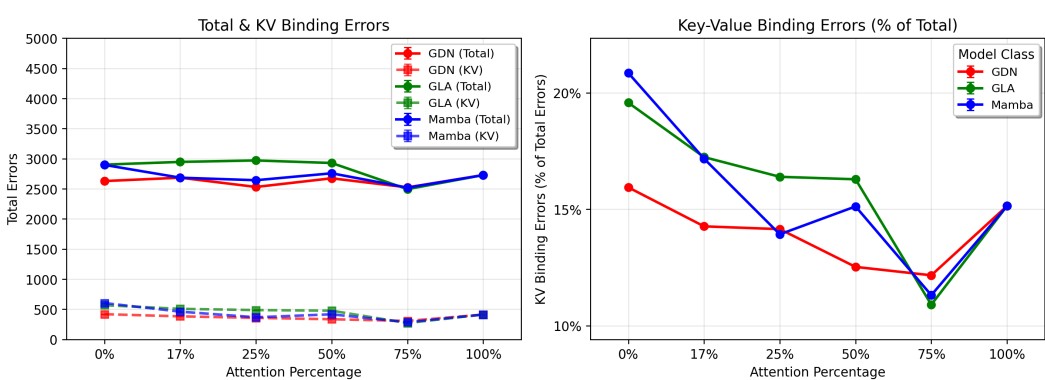

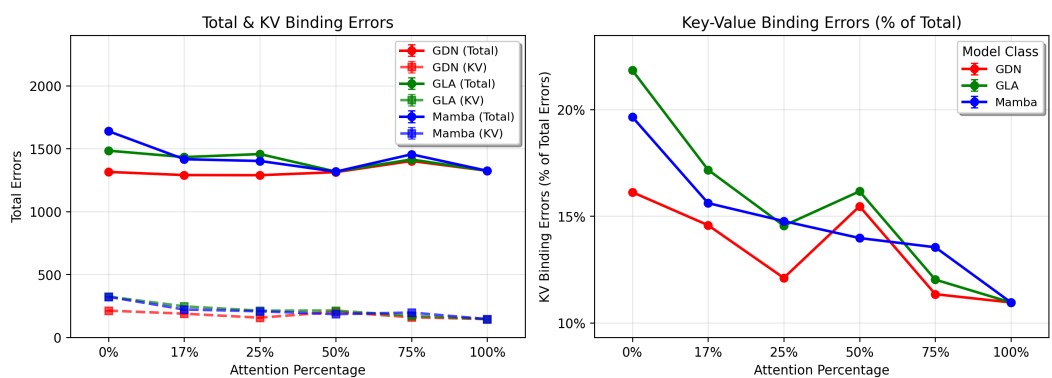

Figure 10: Key–Value binding errors are reduced by attention. **Top:** For 150M models, the total errors and KV-binding errors generally decrease as attention increases across model classes. Note that for the Transformer, there is a slight increase in KV errors as a percentage of the total number of errors, but the general trend downwards holds. Variations might be due to differences in optimization stability of the different architectures **Bottom:** For 500M models, the trend of attention decreasing KV errors is a lot clearer.

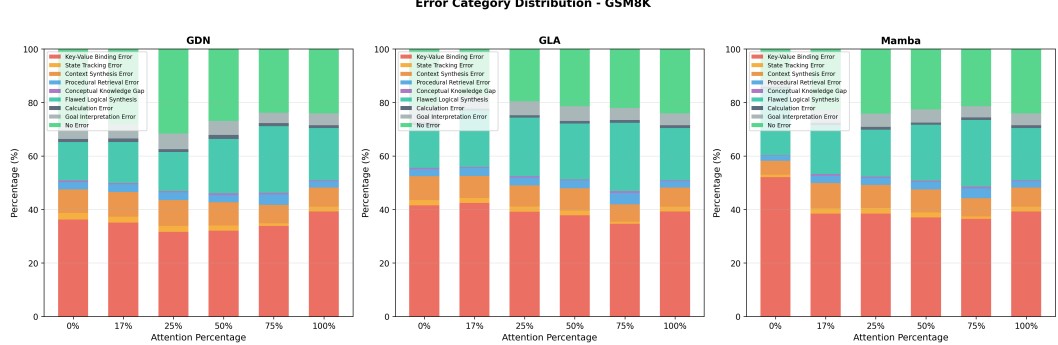

Figure 11: Composite Results of 150M Models on GSM8K showing different error categories as a function of Attention Ratio.

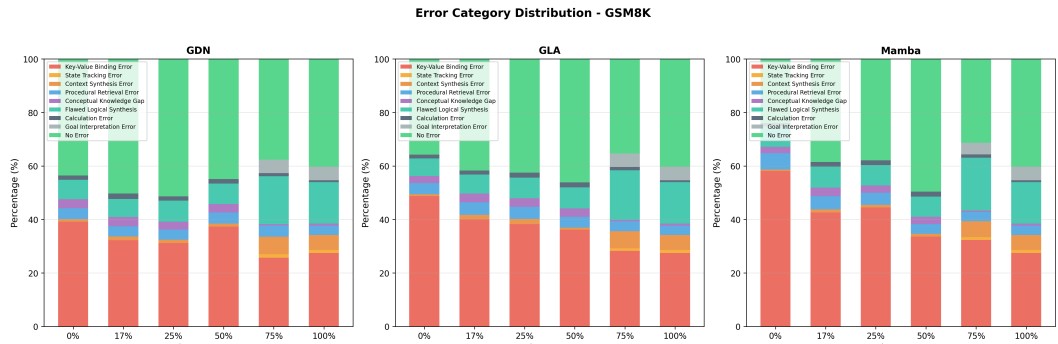

Figure 12: Composite Results of 500M Models on GSM8K showing different error categories as a function of Attention Ratio.

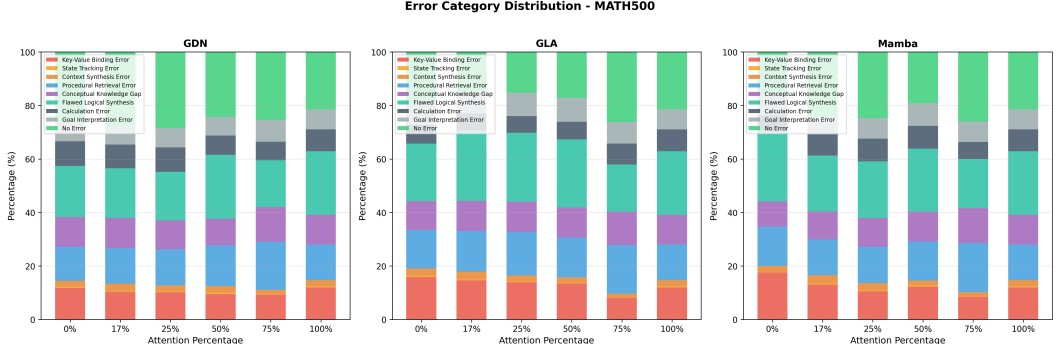

Figure 13: Composite Results of 150M Models on MATH500 showing different error categories as a function of Attention Ratio.

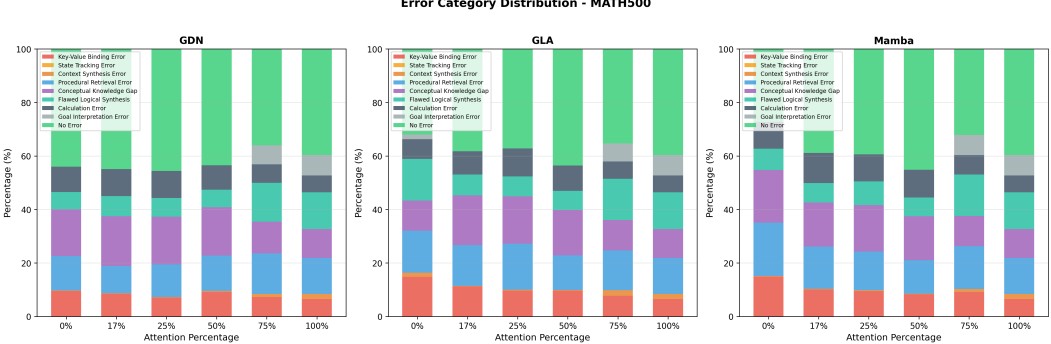

Figure 14: Composite Results of 500M Models on MATH500 showing different error categories as a function of Attention Ratio.

