# OpenReview forum: "From Recall To Reasoning: Understanding the Role of Associative Memory in Hybrid Architectures"
_ICLR.cc/2026/Conference — ICLR 2026 Conference Desk Rejected Submission_

### Official Review · Reviewer_NHaW · 2025-10-28

**Soundness:** 3
**Presentation:** 3
**Contribution:** 3
**Rating:** 6
**Confidence:** 2

**Summary:**

This paper investigates the influence of architectural design on mathematical reasoning by training 150M-parameter hybrid models from scratch, including Mamba, Gated Linear Attention (GLA), and Gated Delta Net (GDN) variants, with attention layer proportions ranging from 0% to 50%. Utilizing a consistent math curriculum (OpenMathInstruct-2 and MetaMathQA), the study assesses performance through parallel test-time scaling (majority voting up to 64 samples) on GSM8K and MATH benchmarks. Results reveal a "reasoning gap," where subquadratic models lag behind Transformers and gain less from additional compute, though performance improves with more attention layers. An LLM-as-a-judge analysis categorizes errors into eight types (e.g., key-value binding, state tracking), identifying associative memory failures as the primary limitation in attention-free architectures, thus linking efficiency trade-offs to reasoning challenges.

**Strengths:**

**Robust experimental control**: Training all models from scratch on a unified dataset removes biases from pre-trained Transformers or distillation, ensuring a rigorous and equitable comparison across hybrid architectures.

**Detailed error taxonomy**: Drawing inspiration from mathematics education, the fine-grained error classification effectively ties subquadratic shortcomings to associative memory deficits, enhancing mechanistic understanding.

**Actionable insights**: The clear "dose-response" relationship between attention layers and reasoning performance offers valuable guidance for hybrid model development.

**Weaknesses:**

For conclusion, the focus on associative memory failures in mathematical reasoning echoes prior recall ability studies [1], suggesting the paper may not significantly advance hybrid model insights beyond established findings.

The observation that some hybrids (e.g., GDN50) achieve competitive pass@1 scores but diverge from Transformers with higher samples (as shown in figures) lacks sufficient explanation, leaving a gap in understanding this critical behavior.



[1] Arora et al. Zoology: Measuring and Improving Recall in Efficient Language Models. arXiv:2312.04927

**Questions:**

Why do certain hybrids (e.g., GDN50) initially outperform Transformers in pass@1 on GSM8K but fall behind with increased samples—could this reflect specific architectural strengths or training artifacts?

How might these results evolve with larger models or non-mathematical domains, and what targeted adjustments (e.g., increased state sizes) could mitigate the associative memory gap without incorporating additional attention?

Is there any best trade-off between efficiency and performance?

---

> ### Author Response · Authors · 2025-11-21
> **Clarifications on Novelty and Architectural Trade-offs (Response to Reviewer NHaW)**
>
> We thank you for the positive review and detailed comments. We hope to address your concerns below.
>
> **1. Novelty Beyond Existing Literature (Addressing Weakness 1)**
> You rightfully point out that this paper has connections to some existing work. However, we want to highlight some key contributions of our paper that extend beyond what is found in the literature and have just recently been validated in industry.
>
> *   **Relating Associative Memory to Reasoning:** While existing work such as Arora et al. identifies performance gaps between Transformers and State Space Models on synthetic associative memory tasks, we are the first to show that this deficiency on in-context associative recall directly contributes to limitations in these models for *reasoning* on real-world tasks. This is an important direction as reasoning models are compute-intensive at inference, and these efficient architectures represent ways to significantly improve model performance.
> *   **Fine-Grained Hybrid Analysis:** By training these models with the same total number of parameters on the same datasets, we control for many different sources of noise in the training procedure. Through this, we are able to better isolate the architectural effects, through which we discover that Gated DeltaNet is highly performant compared to Mamba and GLA. Furthermore, our new GDN ablations show the relative importance of the $\alpha$ and $\beta$ parameters in downstream performance, a novel result which should help inform which architectural components to further design around.
> *   **Test-Time Scaling:** We establish that the attention "dose-response" finding holds along the test-time scaling dimension. This is important, since one of the upsides of a hybrid model is efficient inference, and hence more test-time scaling for the same inference budget. Furthermore, we identify that Transformers generally tend to benefit more from test-time scaling as they are more consistent in their answers.
>
> ***
>
> **2. Why GDN50 starts strong? (Addressing Question 1)**
> You asked why GDN50 sometimes beats Transformers at pass@1.
> Regarding the superior pass@1 performance of GDN50 over pure Transformers, we observe a consistent trend where 50% hybrid architectures outpace full-attention baselines in the 500M parameter Mamba and GLA models (Figure 1 Bottom). We hypothesize that an optimal attention density exists, beyond which additional attention layers provide diminishing returns or even degrade performance. We attribute this to two primary factors.
>
> First, replacing attention layers may mitigate *rank collapse* (or feature over-smoothing), a known issue in deep Transformers (Dong et al., 2021) where representations become indistinguishable across layers; the hybrid layers could potentially serve to "reset" the feature space. Second, the gated layers may act as a natural filter, stabilizing the training signal and handling local dependencies, thereby allowing the remaining attention layers to optimize more effectively for long-range retrieval. We leave as future work an investigation into how the internal representations of the model change to rigorously confirm this hypothesis.
>
> ***
>
> **3. Non-Mathematical Domains and Targeted Adjustments (Addressing Question 2)**
> We focused on mathematics because it offers objective ground truth and established error taxonomies from cognitive science, enabling precise diagnostic analysis that is difficult to replicate in other domains without massive compute. However, because our taxonomy isolates fundamental deficits like state-tracking and key-value binding, we expect these findings to generalize to adjacent fields like code generation.
>
> Regarding adjustments, we added GDN hidden state ablations in Appendix Figure 8. We found that varying the head dimension versus the number of heads (while keeping total width constant) resulted in similar performance. We also added mechanistic ablations (Section 5.2, Table 1) which identify token-dependent decay as the specific mechanism enabling GDN's superior performance.
>
> ***
>
> **4. The Optimal Trade-off (Addressing Question 3)**
> Defining a strict Pareto-optimal architecture is challenging, as it is contingent upon the specific inference stack and kernel optimizations used. However, we empirically observe that GDN 75% represents a favorable sweet spot across scales. It maintains performance competitive with the full-Transformer baseline while delivering significantly higher throughput, effectively balancing generation speed with model quality. This is also supported by recent hybrid model releases from industry, such as **Qwen3-Next** (Qwen Team, 2025) and **Kimi-Linear** (Kimi Team, 2025), which also interleave attention with roughly 75% Gated DeltaNet layers.

---

### Official Review · Reviewer_bLs3 · 2025-10-28

**Soundness:** 4
**Presentation:** 3
**Contribution:** 3
**Rating:** 4
**Confidence:** 3

**Summary:**

This paper examines hybrid-attention model performance on mathematical reasoning tasks, with a focus on the role of attention layers and the models’ ability to do things like key-value binding. The authors analyse the models’ reasoning traces and suggest that poor in-context associative memory is the cause of worse performance of the hybrid models on reasoning tasks relative to a more vanilla transformer, and that including more attention layers tends to improve performance.

**Strengths:**

- Controlled experiment setup, exploring 3 different hybrid architectures and focussing on varying the amount of attention layers. Evidence of solid hyperparameter sweeps to ensure stable training and fair comparisons in Appendix A.
- Analysis of failure cases in reasoning is interesting and reasonable evidence for associative memory being important for reasoning.

**Weaknesses:**

- The MATH results in appendix A are a little concerning: if memory fidelity is quite important, why does the transformer underperform or perform similarly to the Mamba/GDN models for more complicated math tasks? It makes me concerned that the analysis in the main text is fairly specific to GSM8k problems.
- As noted in the paper itself, it would be useful to explore if this is an issue in larger models and/or models that have undergone more pretraining.
- The trend of increasing attention reducing key value binding errors seems a little weak - in figure 3, going from 50->100% attention *increases* key-value binding errors, and in Figure 9, going past 25% attention also seems to not have much effect on the error rate. It would be useful to have more data points here. For GDN, having 100% attention seems to have more errors than 0% attention (figure 3 right)!

While I like the study in this paper, I think that the GDN results seem to push against the hypothesis provided by the authors, with it often bucking the trends of the other models. It would be useful to get more explanations around this.

**Questions:**

- Do you have any idea why GDN seems to not suffer from having 0-100% attention nearly as much as the other hybrid models?
- See weaknesses.

---

> ### Author Response · Authors · 2025-11-21
> **GDN Mechanisms, MATH Evaluation, and 500M Scaling (Response to Reviewer bLs3)**
>
> We thank the reviewer for their detailed and helpful comments. You are correct in observing that Gated DeltaNet (GDN) seems to subvert the trend where increasing attention leads to increased performance. In the paper, we noted that "our results highlight the relative strength of Gated DeltaNet, whose delta-style update rule enables more effective use of finite recurrent memory." To further investigate *why*, we conducted a series of systematic ablations of Gated DeltaNet to identify which architectural components were most relevant to model performance.
>
> **1. Mechanistic Exploration of Gated DeltaNet**
>
> You asked why GDN performs well even with 0% attention, seemingly contradicting the "Attention is necessary" trend. Recent high-performance models from industry, such as Qwen3-Next and Kimi-Linear, adopt GDN/Delta-based backbones for their hybrid models. This prompted us to run a systematic study on the gating mechanisms in GDN. We hypothesize that the *Delta Rule*, which allows for precise overwriting, along with token-dependent $\alpha$ gating, makes GDN particularly robust.
>
> The GDN update rule can be written as:
>
> $$
> S_t = \alpha_t S_{t-1} + \beta_t (v_t - S_{t-1} k_t) k_t^\top
> $$
>
> where $S_t \in \mathbb{R}^{d \times d}$ is the hidden state matrix, and $k_t, v_t \in \mathbb{R}^d$ are the key and value vectors. The term $(v_t - S_{t-1} k_t)$ represents the "prediction error," or the delta between the retrieved value and the target value. Crucially, the update is modulated by two gating factors:
> *   $\alpha_t \in [0, 1]$: The forget gate (decay) controls how much of the previous state is retained.
> *   $\beta_t \in \mathbb{R}$: The step size (write gate) controls the magnitude of the correction to the state.
>
> We conducted experiments setting the gating parameters $\alpha$ and $\beta$ to be either 1, a learnable scalar per head, or a token-dependent parameter per head. Table 1 (included in the updated main text and reproduced below) shows that a token-dependent $\alpha$ is the most critical component of the mechanism.
>
> The ablations reveal a clear hierarchy: configurations with a token-dependent $\alpha$ consistently outperform those with fixed or scalar decay. Regardless of $\beta$ configuration, constraining $\alpha$ to a learned scalar or constant value consistently drops performance. For example, simply switching $\alpha$ from a token-dependent to scalar gate (while keeping $\beta$ token-dependent) drop GSM8K maj@1 performance from 29.0\% to 23.5\%. Interestingly, token-dependent or scalar $\beta$ generally seems to be negligibly better than constant except for when $\alpha$ is token-dependent, in which case changing from scalar or token-dependent $\beta$ to a constant value of $1$ significantly drops performance from $29.9\%$ to $24.5\%$. This also explains why Gated Delta Net significantly outperforms the normal Delta Net architecture (which corresponds in our case to $\alpha=1$ and token-dependent $\beta$). This suggests that for mathematical reasoning, which often requires abilities such as discarding stale variable assignments (e.g., $x=5$) to make room for new ones ($x=10$), the ability to selectively ``forget" via a dynamic $\alpha_t$ is more critical than the precise magnitude of the write operation $\beta_t$.
>
> | Configuration | GSM8K (maj@1) | GSM8K (maj@64) | MATH500 (maj@1) | MATH500 (maj@64) |
> | :--- | :---: | :---: | :---: | :---: |
> | **$\alpha$ (token-dep), $\beta$ (token-dep)** | 29.0 | 50.0 | **27.2** | 43.0 |
> | **$\alpha$ (token-dep), $\beta$ (scalar)** | **29.9** | **50.6** | 25.0 | **44.0** |
> | $\alpha$ (token-dep), $\beta=1$ | 24.5 | 47.9 | 24.2 | 43.4 |
> | $\alpha$ (scalar), $\beta$ (token-dep) | 23.5 | 46.7 | 24.6 | 43.4 |
> | $\alpha$ (scalar), $\beta$ (scalar) | 26.5 | 46.5 | 26.2 | 43.0 |
> | $\alpha$ (scalar), $\beta=1$ | 25.7 | 46.3 | 23.6 | 42.0 |
> | $\alpha=1$, $\beta$ (token-dep) | 21.4 | 47.6 | 23.8 | 39.8 |
> | $\alpha=1$, $\beta$ (scalar) | 23.4 | 46.6 | 21.6 | 40.2 |
> | $\alpha=1$, $\beta=1$ | 23.3 | 46.6 | 22.8 | 40.8 |

---

> ### Author Response · Authors · 2025-11-21
>
> **2. MATH Results (Addressing Weakness 1)**
>
> You pointed out that 150M Transformers underperformed on MATH in the Appendix. We thank you for raising concerns about this result. As we noted, the Transformer initially demonstrated a significant advantage in maj@64 performance over Mamba and GLA, but performed slightly worse than GDN hybrid models.
>
> Upon further investigation, we realized that the generation length of 1024 tokens we used for MATH500 resulted in several truncated answers, skewing the results lower than the models' actual capabilities. This is because the questions in MATH500 (High School Math Competition Problems) are significantly harder than the questions in GSM8K (Grade School Problems) and require longer chains of thought.
>
> We re-ran the experiments with a generation length of 2048. We found that the performance for the Transformer 150M now exceeds the GDN hybrids (Appendix A.1, Figure 4). However, we still note that GDN75 breaks the trend and performs nearly on par with the Transformer, aligning with our broader finding that GDN is a highly performant architecture.
>
> ***
>
> **3. Model Scale (Addressing Weakness 2)**
>
> We trained 500M parameter models on the same dataset mixture and evaluated their performance. We include these results in the main body of the text (Figure 1 Bottom and Figure 3). In particular, we find that while the "Reasoning Gap" persists for Mamba and GLA, 50% hybrid models outperform pure Transformers.
>
> Notably, several recent large models have moved away from ubiquitous full attention, opting instead for sub-quadratic architectures or sliding-window attention. It is important to note, however, that our models were trained on only 37B tokens. Future work will investigate the scaling relationship between model size, token budget, and hybrid ratio.
>
> ***
>
> **4. KV Binding Error Trends (Addressing Weakness 3)**
>
> The reviewer rightfully points out a counterintuitive trend regarding Key-Value (KV) Binding errors. We hope to elucidate this behavior.
>
> Regarding the KV-Binding Errors on MATH500 (formerly Figure 9, now Figure 10 in the updated Appendix), we observed a plateau in the reduction of KV-Binding errors. It is worth noting that we report KV-Binding Errors in the righthand plots as a percentage of total errors, not as a percentage of total responses. This means that if other errors go up or down, that also impacts the relative ratio of KV-Binding errors in this visualization.
>
> Due to the difficulty of the MATH500 dataset, a large portion of the errors are attributable to *Logical & Execution Failures* and *Parametric Memory & Procedural Failures*. Therefore, we hypothesize that the easily reducible KV-Binding errors are already addressed by the time we reach 50% attention. To further investigate this, we are currently running experiments for these different architectures at 75% attention to see if the reduction of associative memory errors plateaus around 50%. We will report these results as soon as they are available.
>
> In the meantime, we added results for the KV Binding Errors on GSM8K and MATH500 for the 500M size models (Figure 3 and Figure 10). In these larger models, we find a clearer trend: increasing attention generally decreases the count of KV errors, despite GDN still tending to perform on par with the Transformer.

---

> > ### Comment · Reviewer_bLs3 · 2025-11-21
> >
> > Thank you for your detailed response! The generation length explanation for the unusual MATH answers makes complete sense, and thank you for the clarifications around why GDN is so strong. I'm happy to update my score based on these clarifications and new results, thank you!

---

### Official Review · Reviewer_ws5E · 2025-11-03

**Soundness:** 2
**Presentation:** 1
**Contribution:** 2
**Rating:** 2
**Confidence:** 3

**Summary:**

For investigating the efficient inference for reasoning (particularly for arithmetic reasoning), the authors test different architectures with different scaling laws for the same size of the models. They analyze the results with the 8 categories depending on the responses across hybrid subquadratic architectures such as Mamba, GLA, GDN versus Transformers, trained on mathematical reasoning datasets.

Tasks requiring in-context recall reveal a consistent performance gap: models without attention layers struggle to maintain associative bindings and state updates, leading to key–value binding and state-tracking errors. The authors argue that attention provides a non-parametric associative memory that recurrent subquadratic models lack. The authors conclude that attention provides a high-fidelity associative memory essential for robust reasoning, whereas subquadratic recurrent models face intrinsic limitations in memory fidelity.

**Strengths:**

1) Motivation is timely and useful.
- Efficient architectures for reasoning (especially Mamba-type SSMs) are the hot topic right now. Asking whether attention is fundamentally required for reasoning is a real and important question.

2) The taxonomy of mathematical reasoning errors are elegant.
- The empirical findings reveal a consistent pattern of in-context associative memory failures across architectures, highlighting a shared limitation in reasoning mechanisms when attention is reduced or removed.

3) The work contributes to the growing understanding of hybrid architectures and their potential in bridging the efficiency–reasoning gap.

**Weaknesses:**

1) The presentation lacks clarity in explaining technical details and experimental designs
- no methodological details to comprehend the contributions, not good enough to replicate the work.
- not sure about the evaluation methods - not quite clear to imagine how its been done? how the LLM-as-a-Judge annotations were validated are under-specified?
- how did you implement and evaluate each categories of reasoning error mechanisms?

2) The applicable domain is too narrowed to arithmetic reasoning, limiting from the general reasoning domains.

3) Empirical studies seem not complete to support the current conclusions/ contributions of the work
- No ablation on training stability, dataset influence, or architectural components beyond attention percentage.

**Questions:**

see weaknesses

---

> ### Author Response · Authors · 2025-11-21
> **Architectural Ablations, Exposition on Methodology, and Reproducibility (Response to Reviewer ws5E)**
>
> We thank the reviewer for their time and detailed comments. We hope to address your concerns. Please let us know if you have any additional concerns.
>
> **1. Architectural Ablations (Addressing Weakness 1)**
>
> You mentioned a lack of ablations on architectural components. We completely agree with the reviewer on the importance of these ablations. We included many ablations in the initial submission and in our revised draft, we have significantly expanded Appendix A.3 to include more details to isolate the factors driving performance. We also added experiments with 500M parameters to explore this phenomenon at a larger scale.
>
> *   **`ShortConv`:** We found that while the specific kernel size of the short convolutional operator doesn't matter, its existence is important (Figure 9, Top Right).
> *   **`Output Gate`:** For Gated DeltaNet (GDN), adding an output gate resulted in slightly worse performance, hence we chose to not include it (Figure 9, Top Left).
> *   **`Layer Placement`:** We observed that keeping the last layer as attention consistently resulted in better performance. We standardized this placement across all architectures for a fair comparison (Figure 9, Bottom).
> *   **`dt_rank`**: Intuitively, this controls the information written to the hidden state in Mamba. We found that a high `dt_rank` resulted in training instabilities (Figure 7, Right).
> *   **`d_state`:** Alongside previous results for Mamba, we include new results for GDN ablating the `state_size` as 32, 64, and 128 (simultaneously reducing `n_heads` to keep model dimension constant) (Figure 8).
>
> **Mechanistic Analysis of GDN Gates**
>
> To explain why GDN outperforms other subquadratic models, we analyzed the update rule:
>
> $$ S_t = \alpha_t S_{t-1} + \beta_t (v_t - S_{t-1} k_t) k_t^\top $$
>
> where $S_t \in \mathbb{R}^{d \times d}$ is the hidden state, and the term $(v_t - S_{t-1} k_t)$ represents the "prediction error." The update is modulated by two factors:
> 1.  $\alpha_t \in [0, 1]$: The forget gate (decay).
> 2.  $\beta_t \in \mathbb{R}$: The step size (write gate).
>
> We conducted experiments setting $\alpha$ and $\beta$ to be either 1, a learnable scalar, or a token-dependent parameter. Table 1 (reproduced below) shows that a token-dependent $\alpha$ is the most critical component.
>
> The ablations reveal a clear hierarchy: configurations with a token-dependent $\alpha$ consistently outperform those with fixed or scalar decay. Regardless of $\beta$ configuration, constraining $\alpha$ to a learned scalar or constant value consistently drops performance. For example, simply switching $\alpha$ from a token-dependent to scalar gate (while keeping $\beta$ token-dependent) drop GSM8K maj@1 performance from 29.0\% to 23.5\%. Interestingly, token-dependent or scalar $\beta$ generally seems to be negligibly better than constant except for when $\alpha$ is token-dependent, in which case changing from scalar or token-dependent $\beta$ to a constant value of $1$ significantly drops performance from $29.9\%$ to $24.5\%$. This also explains why Gated Delta Net significantly outperforms the normal Delta Net architecture (which corresponds in our case to $\alpha=1$ and token-dependent $\beta$). This suggests that for mathematical reasoning, which often requires abilities such as discarding stale variable assignments (e.g., $x=5$) to make room for new ones ($x=10$), the ability to selectively ``forget" via a dynamic $\alpha_t$ is more critical than the precise magnitude of the write operation $\beta_t$.
>
> | Configuration | GSM8K (maj@1) | GSM8K (maj@64) | MATH500 (maj@1) | MATH500 (maj@64) |
> | :--- | :---: | :---: | :---: | :---: |
> | **$\alpha$ (token-dep), $\beta$ (token-dep)** | 29.0 | 50.0 | **27.2** | 43.0 |
> | **$\alpha$ (token-dep), $\beta$ (scalar)** | **29.9** | **50.6** | 25.0 | **44.0** |
> | $\alpha$ (token-dep), $\beta=1$ | 24.5 | 47.9 | 24.2 | 43.4 |
> | $\alpha$ (scalar), $\beta$ (token-dep) | 23.5 | 46.7 | 24.6 | 43.4 |
> | $\alpha$ (scalar), $\beta$ (scalar) | 26.5 | 46.5 | 26.2 | 43.0 |
> | $\alpha$ (scalar), $\beta=1$ | 25.7 | 46.3 | 23.6 | 42.0 |
> | $\alpha=1$, $\beta$ (token-dep) | 21.4 | 47.6 | 23.8 | 39.8 |
> | $\alpha=1$, $\beta$ (scalar) | 23.4 | 46.6 | 21.6 | 40.2 |
> | $\alpha=1$, $\beta=1$ | 23.3 | 46.6 | 22.8 | 40.8 |

---

> ### Author Response · Authors · 2025-11-21
>
> **2. Evaluation Details and LLM-as-a-Judge Methodology (Addressing Weakness 1)**
>
> You asked for more details on the evaluation pipeline and how it is validated. We apologize for any lack of clarity in the initial draft. We have reformatted the paper to pull these details from the Appendix into Section 3.
>
> **The Experimental Pipeline**
> 1.  **Training:** We train models end-to-end on a mixture of OpenMathInstruct-2 (Toshniwal et al., 2024) and MetaMathQA (Yu et al., 2024).
>     *   *Dataset Influence:* You specifically noted a lack of ablation on dataset influence. We initially experimented with other mixtures, but found this specific composition to be the highest performing. This finding was independently corroborated by the extensive dataset ablations in Zhao et al. (2025), which identified this specific mixture (OMI2 + MMQA) as the optimal initialization for reasoning.
> 2.  **Generation:** We generate 64 responses per question on two datasets: GSM8K and MATH500.
> 3.  **Test-Time Scaling:** We use "majority voting" (self-consistency) to identify the most common response per question. We evaluate performance as we increase the number of responses by powers of 2 (from $N=1$ to $64$).
>
> **LLM-as-a-Judge Diagnostic & Validation**
>
> While scaling curves show *that* models fail, they do not explain *why*. We introduce an LLM-as-a-Judge methodology to diagnose failure modes at scale:
> *   **Judge Model:** We use Gemini-2.5 Flash-Lite. We provide the full system prompt used for evaluation in Appendix C.1, which includes the strict rubric and JSON schema. Additionally, Appendix C.3 contains concrete examples of reasoning traces for every error category to illustrate the taxonomy in practice.
> *   **Validation:** You specifically asked how these annotations were validated. Beyond citing established literature (see below), we performed qualitative manual inspection on a subset of the judge's outputs to verify the robustness of the taxonomy. We found high agreement between human assessment and the judge's categorization. Through structured prompting and strict JSON output enforcement, the judge analyzes each answer independently to avoid bias.
>
> **Error Taxonomy**
>
> To ensure meaningful insights, we utilize a specialized 8-category taxonomy grounded in cognitive science:
> *   **In-Context Associative Memory (ICAM) Failures:** Issues with reading variables (Key-Value Binding) and updating them (State Tracking).
> *   **Procedural Failures:** Internal knowledge gaps or faulty algorithms.
> *   **Logical Execution Failures:** Flaws in strategic planning or calculation.
>
> This approach allows us to discover that Key-Value Binding errors make up a significant portion of failures in subquadratic models, and that increasing Attention density directly reduces these errors.
>
> **References**
>
> We have added the following references to the main body to contextualize this method:
> *   *Judging LLM-as-a-Judge with MT-Bench* (Zheng et al., 2023)
> *   *AlpacaEval* (Li et al., 2023)
> *   *G-EVAL* (Liu et al. 2023)
> *   *A Survey on LLM-as-a-Judge* (Gu et al. 2025)
>
> ***
>
> **3. Limitations to Arithmetic Reasoning (Addressing Weakness 2)**
>
> You noted that the domain is narrowed to arithmetic. While true, we believe this is a necessary constraint for a rigorous study of reasoning mechanics:
> 1.  **Objective Verification:** Unlike general chat, math problems have objectively verifiable final answers, allowing us to accurately measure the efficacy of test-time scaling.
> 2.  **Established Diagnosis:** We can draw on decades of math education literature to categorize errors (e.g., "State Tracking" vs. "Calculation Error"). This depth of analysis is difficult to replicate in other domains without established diagnostic frameworks.
> 3.  **Foundation Model Training:** Since foundation models are trained on large amounts of math data, the Judge can effectively isolate reasoning errors from simple hallucinations since it has experienced with this type of data.
>
> However, we believe that core memory deficits we identify such as Key-Value Binding are fundamental to sequence modeling and likely generalize to reasoning in other state-heavy tasks like code generation or long-document analysis.
>
> ***
>
> **4. Reproducibility (Addressing Weakness 3)**
>
> We have expanded the manuscript to ensure full reproducibility. We provide specific details in the following sections:
>
> *   **Appendix B.1:** Hyperparameters specific to model architectures (GDN, Mamba, GLA).
> *   **Appendix B.2:** Training dataset composition and curriculum details.
> *   **Appendix B.3:** Training hyperparameters (learning rates, optimizers, schedules).
> *   **Appendix B.4:** Test-time scaling implementation details.
>
> Our setup is built on the open-source OLMo2 and flash-linear-attention libraries. All datasets used are hosted on HuggingFace. We plan on fully releasing the entire codebase—including the training protocol, inference scripts, LLM-as-a-Judge pipeline, and visualization scripts—upon publication.

---

### Author Response · Authors · 2025-11-21
**General Summary**

# Summary of Major Updates: 500M Scaling, Mechanistic Analysis of Gated DeltaNet, and Improved Exposition

We thank the reviewers for their detailed and helpful feedback. We have uploaded a revised manuscript with a significantly expanded experimental suite and clearer explanation of our methodology. In addition, we obtain novel results which reinforce our core finding: while associative memory remains a bottleneck for pure SSMs, some hybrid architectures such as Gated DeltaNet (GDN) can effectively bridge this gap. Notably, we provide a mechanistic explanation into the relevant components of the GDN architecture and comment on how our best performing models match recent findings in large-scale hybrid models from industry (e.g., Qwen3-Next and Kimi-Linear).

Below, we summarize the four major updates to the paper:

### 1. Scaling to Larger Models (500M)
Reviewers asked if the "reasoning gap" holds at larger scales. We trained and evaluated 500M parameter versions of all architectures (Transformer, Mamba, GLA, GDN) on the same curriculum. We include these results in the main body (Figure 1 Bottom) and find:
*   **The Reasoning Gap:** The gap persists for pure Mamba and GLA models, which leverage test-time compute less effectively than Transformers.
*   **Hybrid Efficiency:** 50% Hybrid models (particularly GDN) at 500M scale are able to match or slightly exceed the Transformer baseline.
*   **Error Analysis:** We confirm that KV-Binding Errors generally decrease as Attention percentage increases (Figure 3 Bottom), reinforcing our hypothesis that attention acts as a high-fidelity associative memory.

### 2. Mechanistic Analysis of Gated DeltaNet
Reviewers noted that GDN outperforms the general trend, performing well even with low attention. To explain this, we conducted targeted ablations on the GDN layer (Table 1). We found that **token-dependent decay ($\alpha$)** in the GDN update rule is the most critical component. When the $\alpha$ mechanism is disabled or restricted to scalar decay, GDN's performance reverts to having a higher error rate.

This confirms that while Mamba struggles with information decay, GDN succeeds because it can selectively "overwrite" specific associations. This provides an independent scientific explanation for the design choices of very recent models like Qwen3-Next and Kimi-Linear, which both converge to an optimal ratio of 75% Gated DeltaNet.

### 3. Correction to MATH500 Methodology
In response to valid concerns regarding the Transformer's underperformance on the MATH dataset in our initial appendix, we reinvestigated our evaluation pipeline with different configurations. We discovered that the generation length (1024 tokens) was insufficient for the complex reasoning required by MATH500, leading to premature truncation. Instead, we re-ran these experiments with a generation length of 2048 (Appendix A.1).

The Transformer now outperforms the Mamba/GLA models as expected, resolving the anomaly. However, GDN hybrids remain highly competitive, further validating the strength of the Delta rule.

### 4. Exposition & Reproducibility
We have significantly expanded Section 3 and Appendix C to detail our dataset composition and LLM-as-a-Judge pipeline. We provide clear examples of the 8 distinct error modes identified in real model answers. We emphasize that we will release the full codebase—including the dataset, training protocol, inference scripts, and the automated error analysis pipeline—upon publication to ensure full reproducibility.

### 5. Final Confirmatory Experiments
To complete the sweep of determining the optimal hybrid ratio, we are currently finalizing runs for GDN, GLA, and Mamba with 75% Attention (Mamba25, GLA25, GDN25) at both 150M and 500M scales. While these architecture ratios generally would not be used in practice as they would provide very little efficiency gains, we want to explore them for more fine-grained analysis. Preliminary results align with our existing trends, and we will update the paper with the final data points as soon as the runs complete. (Update: We finished and added these results to the paper).

---

### Note · Program_Chairs · 2026-01-13
**Submission Desk Rejected by Program Chairs**

The original submission of this paper includes language that deanonymizes the authors and consequently must be desk rejected.